# Drone-based meteorological observations up to the tropopause – a concept study

Konrad Bärfuss[1], Holger Schmithüsen[2], and Astrid Lampert[1]

[1]Technische Universität Braunschweig, Institute of Flight Guidance, 38108 Braunschweig, Germany
[2]Alfred Wegener Institute, Helmholtz Centre for Polar and Marine Research, 27570 Bremerhaven, Germany

**Correspondence:** Konrad Bärfuss (k.baerfuss@tu-braunschweig.de)

**Abstract.** The main in-situ database for numerical weather prediction currently relies on radiosonde and airliner observations, with large systematic data gaps, horizontally in certain countries, above the oceans and in polar regions, and vertically in the rapidly changing atmospheric boundary layer, but also up to the tropopause in areas with low air traffic. These gaps might be patched by measurements with drones. They provide a significant improvement towards environment friendly additional data, avoiding waste and without the need for helium. So far, such systems have not been regarded as a feasible alternative of performing measurements up to the upper troposphere. In this article, the development of a drone system that is capable of sounding the atmosphere up to an altitude of 10 km with own propulsion is presented, for which Antarctic and mid-European ambient conditions were taken into account: After an assessment of the environmental conditions at two exemplary radiosounding sites, the design of the system and the instrumentation are presented. Further, the process to get permissions for such flight tests even in the densely populated continent of Europe is discussed, and methods to compare drone and radiosonde data for quality assessment are presented. The main result is the technical achievement demonstrating the feasibility of reaching an altitude of 10 km with a small meteorologically equipped drone using its own propulsion. The first data are compared to radiosonde measurements, demonstrating an accuracy comparable to other aircraft based observations, despite the simplistic sensor package deployed. A detailed error discussion is performed. The article closes with an outlook on the potential use of drones for filling data gaps in the troposphere.

## 1 Introduction

Accurate weather predictions are of high importance for humankind, from agriculture via air traffic, warning of severe weather events like storm and heavy rain to the personal activities of individuals. With increasing computational power, there have been significant improvements in operational weather models (Bauer et al., 2015). However, these global and mesoscale models require measurement data as input to tie the short-term forecast towards observations (Wang et al., 2000). In this computing-intensive process, data can be assimilated continuously, with high flexibility regarding spatial and temporal resolution and trajectory (Bonavita et al., 2016). The data to be assimilated originate from the World Meteorological Organization (WMO) Global Observing System (Thépaut and Andersson, 2010; WMO, 2010, 2015a), consisting of measurements using both in-situ and remote sensing techniques. Atmospheric measurements of pressure, temperature, humidity, wind speed and wind direction

are crucial to Numerical Weather Prediction (NWP). These measurements can partially be provided by ground based remote sensing techniques (Lindskog et al., 2004; Kotthaus et al., 2022), satellite based remote sensing techniques (Steiner et al., 2001; Karbou et al., 2005; Rennie et al., 2021), radiosondes (Ingleby et al., 2016a), aircraft (Fleming, 1996) and drop sondes (meteorological sensor packets dropped from high altitude platforms (Hock and Franklin, 1999). Each of these observing system types has its own peculiarities which have to be considered for implementing in weather models, and has a different impact on the forecast quality.

Ground based remote sensing instruments need significant financial effort to be deployed and operated. Their use around the globe is therefore quite limited. Satellite based remote sensing measurements provide superior global coverage and have a high impact in NWP (Cardinali, 2009), especially over data-poor areas (Bouttier and Kelly, 2001). For calibration and validation of these satellite sensors and data products, satellite based observing systems (and in general remote sensing measurements) rely on in-situ data for calibration and validation (Goldberg et al., 2011; Chander et al., 2013; Boylan et al., 2015; Carminati et al., 2019). An increasing challenge (and also a potential opportunity, Palmer et al., 2021) for retrieving meteorological observations from satellite microwave instruments (e.g. Karbou et al., 2005) is the "society's insatiable need for the radio spectrum." — (Palmer et al., 2021), potentially harming future measurements. Space-based Doppler Wind Lidar measurements are regarded as essential data for weather models, addressing the urgent need in providing wind profiles at all latitudes and altitudes (Baker et al., 2014).

Of major importance regarding in-situ observations are vertical profiles measured by radiosondes. They are launched at specific stations and at fixed launch times, from typically daily to 4 times per day. The measurement data are transferred to the ground via telemetry, and are sent to the network Global Telecommunication System (GTS) to be accessed by weather services in a specific data format (Ingleby and Edwards, 2015). Typical state of the art sounding systems provide measurements of altitude, pressure, temperature, humidity, wind speed and wind direction once per second. Depending on the sounding system and balloon sizes used, radiosondes typically measure atmospheric profiles up to an altitude of 35 km or even 40 km, covering the entire troposphere and most of the stratosphere. Usually, radiosondes are not collected and re-used, but remain in the landscape as litter. The worldwide around 800 radiosonde launch sites are not evenly distributed around the globe. There are large areas with only few regular launches, in particular above the oceans and in the polar regions.

Another important source of in-situ data originates from the AMDAR (Aircraft Meteorological DAta Relay) and the U.S. related TAMDAR (Tropospheric Airborne Meteorological Data Reporting) programme (Moninger et al., 2003; Petersen et al., 2015, 2016; Petersen, 2016). In the vicinity of large cities, vertical profiles of temperature, wind speed and wind direction (and partly humidity) are measured frequently through commercial aircraft equipped with the AMDAR specific meteorological sensor package (WMO, 2003), with additional observations at flight level provided during cruise. For this airborne method, a careful calibration and processing of the data is required (de Haan et al., 2022). Due to less coverage with en route flights and especially airports, regions like the Arctic and Antarctic as well as mid-Africa suffer from a lower data density regarding the AMDAR system.

Data comparable to aircraft and radiosonde measurements can be gathered using sondes dropped by aircraft – either crewed (Laursen et al., 2006) or uncrewed (Kren et al., 2018) - and balloons (Wang et al., 2013) from altitudes close to the ground

up to 30 km (Cohn et al., 2013). Data from drop sonde measurements have recently been used for intense observation periods (Redelsperger et al., 2006; Rabier et al., 2010; Kren et al., 2018; Schindler et al., 2020; Ralph et al., 2020; Zheng et al., 2021), but as they are used for specific target areas and specific purposes only, they do not play a significant role in global observations. To assess the impact on forecast quality and subsequently the importance of these different observing systems (single observation systems) in order to further develop the Global Observing System, Observation System Experiments (Bormann et al., 2019), Sensitivity to Observation (Cardinali, 2013; Langland and Baker, 2004) and similar experiments (Ingleby et al., 2020; Ota et al., 2013) can be carried out. As new observing systems are deployed, their impact on weather models (including the assimilation system) is evaluated and reviewed using these techniques (Rennie et al., 2021; Petersen, 2016; Petersen et al., 2016; Bouttier and Kelly, 2001; Bormann et al., 2019; Riishojgaard, 2015). For example, space-based Doppler Wind Lidar measurements are regarded as essential for numerical weather prediction (Baker et al., 2014). The examination of including wind retrievals using the first space-borne wind Lidar showed a positive impact on forecasting quality (Rennie et al., 2021). Aircraft meteorological measurements complement radiosonde measurements when radiosonde data were not used in the forecast experiments (Gelaro and Zhu, 2009; Lorenc and Marriott, 2014; Kim and Kim, 2019). Aircraft wind and temperature reports show a significant improvement of model results at pressure levels between 700 - 400 hPa (around 3-7 km altitude) (Petersen, 2016). The availability of additional aircraft humidity data have the highest impact between 1000 - 400 hPa (around 0.5-7 km altitude) (Petersen et al., 2016), whereas additional radiosonde in-situ humidity data have the highest influence on weather models at 700 - 600 hPa (around 3-4 km altitude) (Ota et al., 2013). In comparison with temperature and wind, the impact of aircraft humidity data showed lower influence on the results of weather models (Ingleby et al., 2020).

Summing up the components of the Global Observation System, in-situ data gaps of important observations are obvious, as radiosonde and aircraft based soundings are sparsely distributed over remote areas and oceans, associated with increased impact of additional radiosonde observations (Ota et al., 2013). The density of observations is not well-balanced with the user requirements for observations. Breakthrough requirements as defined in WMO (2015b) from data users exceed today's capabilities of the Global Observation System in terms of temporal and spatial resolution for the use case of global and high resolution numerical weather prediction. These requirements differ between their application area and the variable, e.g., for global NWP, the breakthrough requirement for the spatial resolution of temperature measurements is 100 km horizontally, 1 km vertically and an observation cycle of 6 h with a timeliness of 30 min (Leuenberger et al., 2020). Generally spoken, more and higher resolution data lead to improved numerical simulations of both local and regional weather forecast (Faccani et al., 2009). Numerical weather prediction models perform best with observations of similar temporal and spatial resolution as in the model (Dabberdt et al., 2005).

Besides regional data gaps (Ingleby et al., 2016b; WMO, 2020) and general data gaps (Houston et al., 2021), there is a data gap in the lower troposphere in atmospheric observing systems (Leuenberger et al., 2020; Pinto et al., 2021), and the potential of drones to fill the gap is currently being discussed. Drones (also called unmanned or uncrewed (Joyce et al., 2021) aircraft systems, UAS, or remotely piloted aircraft systems, RPAS) provide a flexible tool for atmospheric sensing. The use of small drones as a platform for meteorological sensors dates back to the early 1960s (Konrad et al., 1970). Far from being mature at that time, their use was limited to augmented line of sight operations using binoculars to monitor the aircraft's attitude and

therefore the use in the lower troposphere. A comprehensive review of the historical and recent use of fixed-wing drones for meteorological sensing can be found in Elston et al. (2015). In consequence of the emergence of commercial off-the-shelf drones (both fixed-wing and multicopter), the use of drones in different fields of research rapidly increased during the last decade.

The atmospheric boundary layer experiences high temporal changes, and as being closely connected to the spatially variable Earth surface, the boundary layer plays a key role in initiating or hindering weather events like convection or cloud and fog formation. Therefore, drone measurements in the boundary layer have a high potential for providing data of added value to weather forecast (Inoue and Sato, 2022), for example by determining the boundary layer altitude capped by a temperature inversion (Jonassen et al., 2012; Flagg et al., 2018).

The improvement of assimilating drone measurements of the atmospheric boundary layer in numerical weather predictions during intensive meteorological campaigns has been demonstrated (de Boer et al., 2020), with improvements of modelling results for a distance up to 300 km (Sun et al., 2020). Significant benefit from regular drone soundings even to limited altitudes of 1 km or 3 km has been demonstrated for precipitation (Chilson et al., 2019) and cloud coverage (Leuenberger et al., 2020), and a reduction of over 40 % for the root-mean-square error and bias in the 15-min forecasts of temperature, wind, and humidity between the benchmark run and a model run with assimilated data of a coordinated fleet of drones was observed (Jensen et al., 2021, 2022), despite the challenges of data assimilation in mountainous environments (Hacker et al., 2018). However, drone measurements up to higher altitudes would be more beneficial (Sun et al., 2020). It must be noted here, that the deployment of drone systems for operational meteorology only has benefits, if data can be transferred and distributed in near real-time, which has not been demonstrated within most of the above-mentioned studies.

For obtaining additional data similar to the classical radiosondes, balloon-launched drones which are carried up to a certain altitude, which can be around 20-40 km, have been developed. Once reaching the target altitude, the systems are released from the balloons and then return to the starting location in restricted airspace (Lafon et al., 2014; Kräuchi and Philipona, 2016; Schuyler et al., 2019) - at least for low wind speed conditions. These drones further provide the advantage of controlling the direction of flight. In comparison to radiosondes, it is therefore possible to deploy more sophisticated instrumentation, as it can be used multiple times, and sensors can be calibrated before and after a sounding for quality checks. High data quality enables further use of the measurements, such as climate applications (see ANNEX 12.B in WMO, 2018). However, balloon launched systems require the availability of helium and a certain launching infrastructure, like for the classical radiosonde launches, and are barely able to glide back to their starting location for wind speed exceeding $15 \, \mathrm{m \, s^{-1}}$.

For increasing the flexibility of the launching site, it is beneficial to deploy systems with own propulsion. Regular soundings of multicopter drones to improve weather forecast for airports have been established in Switzerland (Leuenberger et al., 2020). Other studies to improve weather prediction include fixed-wing drones as well (Koch et al., 2018). Further, no waste is left from such a drone ascent, which is of high importance in particular in the Antarctic, where the Antarctic Treaty requires Environmental Impact Assessments to be developed for all activities and which sets rules for waste disposal and management (Secretariat of the Antarctic Treaty, 2019). Nevertheless, systems with own propulsion normally do not reach the altitudes of radiosondes, and therefore can be compared more easily to aircraft (e.g. AMDAR/TAMDAR). The main technical challenges

of drone operations up to high altitudes comparable observations on commercial aircraft are high wind speed, low temperatures, potential icing and extremely low specific humidities regarding measurement techniques. Individual systems and concepts have been developed for applications in high wind speed, like for in-situ measurements of hurricanes and tornadic supercells (Cione et al., 2020; Elston et al., 2011) and for measurements up to and within the stratosphere (Runge et al., 2007).

Nowadays, the status of small drones for weather sensing in the lower troposphere is quite mature (Pinto et al., 2021) and close to being ready for operational applications in meteorology. As an experiment to collect experience on both the drone operation and the NWP side, the WMO is preparing a worldwide coordinated demonstration campaign in 2024 (WMO, 2022). There is broad agreement that drones are becoming an increasingly important tool for all sorts of meteorological tasks (Geerts et al., 2018; Vömel et al., 2018). Interestingly, drones which are capable of reaching the upper troposphere and even lower stratosphere are viewed as a future alternative to current in-situ observing systems and enable the community to address important scientific issues, but these drone systems did not receive much attention in the science discussion – the road to inexpensive high-flying drones seemed to remain unpaved.

The importance of aircraft measurements in the troposphere concomitant with in-situ data gaps in the troposphere and over remote areas was the starting point for a research project at the Technische Universität Braunschweig (Germany), in which a drone was developed to augment in-situ data in Antarctica. The drone represents a fairly unusual class (medium altitude, short endurance, Watts et al., 2012). The propelled drone technique presented here provides the capability of sounding the entire troposphere vertically or perform level legs at designated altitudes while measuring pressure, temperature, humidity, wind speed, wind direction and turbulence, and flexibly addressing the data needs (Houston et al., 2021). Regarding data assimilation, drone data transferred to the GTS is similar to aircraft data (especially true for regional aircraft observations, Moninger et al., 2010) and radiosonde data (ascent and descent).

This feasibility study presents the concept, design, and first applications of the system *LUCA* (Lightweight Unmanned high Ceiling Aerial system), which was developed to provide complementary in-situ data up to an altitude of 10 km at flexible locations. Simultaneous radiosonde ascents are used to validate the quality of the meteorological observations acquired during the first flights.

Please be aware of Appendix A - D to which the authors paid particular attention. Appendix A contains a detailed description of the data processing, Appendix B introduces generally valid calibration techniques, Appendix C theoretically estimates measurement errors and Appendix D presents a note on the variability of the atmosphere.

## 2 Methods

In the following, the process towards the design of the drone of type *LUCA* is presented briefly. Requirements for the system are derived from the environmental conditions to be expected, to obtain a high availability of measurements. Based on the environmental conditions for two sites, the design of the mission and of the drone is introduced. The simplistic sensor package that was used for the demonstration flights is described, including uncertainty aspects. The process of obtaining flight permissions

for such altitudes is presented. The methods for data post-processing are presented, and the obtained data quality is assessed. The section closes with a note on the variability of the atmosphere.

## 2.1 Environmental conditions

*LUCA* was designed to operate in mid-latitude and polar conditions. Therefore, the expected environmental constraints were evaluated from the radiosonde stations Neumayer in Antarctica and Lindenberg in Germany (Figure 1) episodically over three years (2016-2018). The temperature range covering 90% of the operation conditions of the ascents at Lindenberg is between -60°C and 20°C and at Neumayer between -75°C and -2°C. For the station Lindenberg, the median wind speed is up to $20\,\mathrm{m\,s^{-1}}$, and the wind speed that is encountered in 90% of the cases (90% percentile) is up to $42\,\mathrm{m\,s^{-1}}$ for the altitude of the jet stream (7-15 km) (Pena-Ortiz et al., 2013). For the station Neumayer, the median wind speed is up to $14\,\mathrm{m\,s^{-1}}$, and the wind speed that can be expected in 90% of the cases is up to $28\,\mathrm{m\,s^{-1}}$.

Operational challenges at Neumayer arise from the surface wind speed, which is generally higher than in Europe. The most frequent surface wind conditions are either from East due to cyclonic activities near the polar front, with typical wind speed of $20\,\mathrm{m\,s^{-1}}$, or from S due to katabatic flow, with typical wind speed of $10\,\mathrm{m\,s^{-1}}$ (König-Langlo et al., 1998). High wind speed values are further reached at the altitude of the tropopause, around 9-13 km (Evtushevsky et al., 2008) in the polar jet stream (Archer and Caldeira, 2008).

The system *LUCA* was designed for operation in the temperature range between -75°C and +30°C, and for a wind speed of less than $28\,\mathrm{m\,s^{-1}}$ over the whole vertical profile to limit the maximum vector displacement of the drone from its launch site. Assuming the system is capable of operating in conditions exceeding the design temperature and nominal wind speed by 15 %, this would allow ascents in 87 % of the radiosonde days at Neumayer in the Antarctic and 72 % at Lindenberg. Restricting wind speed conditions such that the drone does not have to stay above the launch point, but must be able to return to the base, the mean wind speed over the atmospheric profile to be observed by the drone should not exceed the nominal horizontal airspeed component of $28\,m\,s^{-1}$. Despite the limited applicability of this approach when using the drone in a reserved airspace and ignoring possible environmental threats such as heavy precipitation or in-flight icing, the system is capable of covering 94 % of the measurements in Lindenberg and 98 % at the Neumayer Station. Adding the margin of 15 %, the availability increases to 97 % for Lindenberg, and 100 % for Neumayer. At Neumayer Station, typically on 96% of the days, radiosondes can be launched. Lindenberg has a temporal coverage of 99% of all days. In addition, the development of the drone addresses measurements during rainfall, snow, heavy turbulence and within clouds, but these capabilities have yet to be proven in upcoming measurement campaigns.

Particular attention was paid to takeoff and landing under high surface wind conditions, which are accompanied at the Neumayer Station by low visibility due to drifting and blowing snow. The probability of 23 % to operate under conditions with a visibility below 500 m requires a highly automated take-off and landing procedure, which does not rely on visual contact of the operator with the system, similar to operations shown by Reineman et al. (2016).

A possible threat for drone measurements are in-flight icing conditions, which depend on temperature, humidity, and droplet size (Jeck, 2002). An idea of the frequency of icing conditions might be available of icing forecast data using ADWICE (Ad-

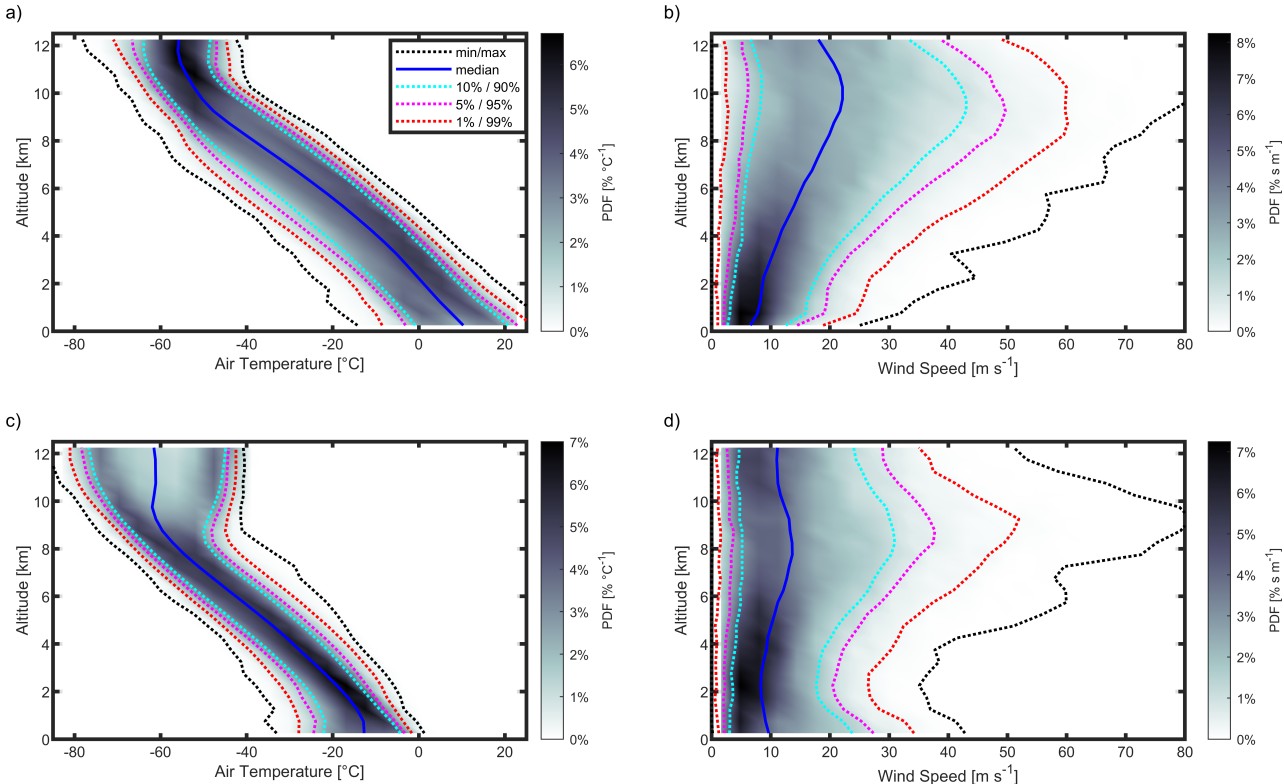

**Figure 1.** Analysis of environmental constraints at the radiosonde stations Lindenberg in Germany (upper panels) and Neumayer in the Antarctic (lower panels). The left panels show the probability distribution function (PDF) of air temperature with altitude in gray scale, the right panels the PDF of wind speed with altitude. The median is indicated in blue, the minimum and maximum values as black dotted line, and the percentiles including 80% (light blue), 90% (magenta) and 98% (red) of the data are indicated. The plot is based on the time period from the year 2016 to 2018, including all radiosonde launches at Neumayer around 12 UTC, and all launches around 00 UTC as well as 12 UTC at Lindenberg.

vanced Diagnosis and Warning system for aircraft Icing Environments, Tafferner et al., 2003) along with validation studies using PIREPs (PIlot REPorts). Such comparisons exist for regions with dense air traffic, e.g., Europe (Kalinka et al., 2017), but no validation studies are found for the Antarctic. In addition, icing differs strongly between manned aircraft (what ADWICE is made for) and drones (Hann, 2020). A report for Norway and surrounding regions explicitly focused on icing for drones using meteorological reanalysis data (ECMWF ERA5, Hersbach et al., 2020) and an ice accretion model (ICE3D, Sørensen et al., 2021), and found theoretical icing frequencies of 45 % at an altitude between 1 km - 1.5 km between September and May, and a lower risk with the highest frequency of 30 % peaking at 2.5 km altitude in June to August. These values are not directly applicable to the Antarctic, but the process to determine the likelihood of icing might be used to preliminarily estimate icing frequency and icing risk for drones in the Antarctic.

Drones are usually operated without sophisticated anti-icing and de-icing concepts like in manned aviation, in particular as

most drones are operated in visual line of sight. However, icing is a threat that may lead to a complete loss of the system. Icing protection for drones has been demonstrated (Hann et al., 2021), but requires additional substantial energy for heating, even if combined with specific ice phobic coatings or liquids (Huang et al., 2019).

For the demonstration flights shown here, the drone *LUCA* was prepared to be equipped with an icing sensor to measure in situations with a substantial risk of meeting icing conditions during the flight but not installed during the demonstration flights as zero risk of icing was present. Together with monitoring performance parameters, this allows estimating the severity of icing and supports the decision of abandoning the mission in icing cases. More details about the all-weather strategy is found in Bärfuss et al. (2022).

## 2.2 System design

Before designing the physical aircraft system, the mission to be accomplished by the drone was defined. Although data assimilation is nowadays highly flexible regarding time, the mission was designed according to radiosonde observations to hypothetically surpass the 100 hPa surface at 12 UTC (Ingleby et al., 2022) and ensure timeliness in foresight of future in-flight reporting analogue to radiosonde reporting, where a first dataset is sent to the GTS when the sonde reaches the 100 hPa level (Ingleby and Edwards, 2015). With a targeted climb rate of $10\,\mathrm{m\,s^{-1}}$ arbitrarily chosen from simple analytical estimates, the drone has to be launched around 11:33 UTC, and reaches the design ceiling of 10 km at 11:50 UTC. During the flight, 1 Hz real time data are available. After descending with a vertical rate similar to the climb rate, an approach procedure is flown in the vicinity of the landing site to determine wind direction and wind speed, and subsequently the approach trajectory is calculated automatically on the onboard computer. After the "splashdown" into a horizontal landing net, the drone can be recovered, and data processing including quality checks and transcoding begins. The observations of the complete flight are finally transferred into the GTS around 13 UTC. Thus, post-processed data from the descent profile are targeted to be available within less than an hour after the measurements are taken. Figure 2 illustrates the mission design.

As key design driving parameters, the ability to fly against high wind speed, an efficient electric propulsion chain and a highly flight-state independent position for the sensor package are essential. These requirements led to the development of a tailless fixed-wing configuration with a pusher propeller. As the result of a multi-variant optimisation for profiling the atmosphere vertically up to 10 km, the design weighs 5-6 kg, depending on the deployed sensor package, has a wingspan less than 2 m, and operates at a constant airspeed of $30\,\mathrm{m\,s^{-1}}$ with a typical ascent rate of $10\,\mathrm{m\,s^{-1}}$ controlled by the autopilot system. Thus, the system has a minimum airspeed of $18\,\mathrm{m\,s^{-1}}$, exceeding the minimum airspeed of crewed ultralight aircraft. Subsequently, it is not feasible to launch the drone by hand, and a dedicated mechanical catapult has been adapted and used during the measurement campaign (Figure 3a). As automatic take-off and landing is required for future operations in zero visibility conditions, a horizontal landing net has been developed and an appropriate manoeuvre to get the drone safely into the net was implemented in the autopilot firmware (ArduPilot). The manoeuvre itself can be considered as an automated vertical dive into the horizontally arranged net. Whereas a belly landing is principally possible, the net landing technique was preferred as it protects the drone and the sensors from the hostile surface in the Antarctic, and prevents the system to be blown away

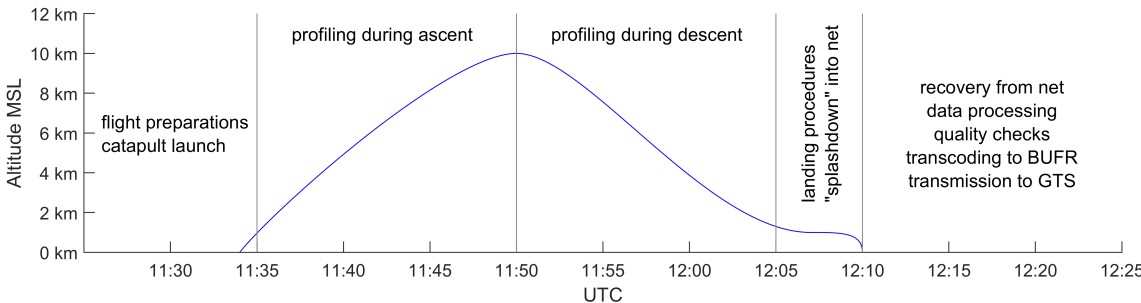

**Figure 2.** Mission design for the drone *LUCA*. In order to be able to provide a first observation data set for assimilation at 12:00 UTC, the drone is launched from the catapult at 11:33 (as shown in Figure 3a) and starts with the vertical sounding in the form of spirals at 11:35, in order to hypothetically cut the 100 hPa level at 12:00 UTC. After reaching its target altitude of 10 km at 11:50, the drone descends with a sink rate equally to the rate of climb ($10\,\mathrm{m\,s^{-1}}$) until it arrives at the designated approach altitude around 12:05. After circles to determine the speed and direction of the near-surface wind, *LUCA* begins with the approach and lands in a horizontal landing net. Data processing, quality checks and transcoding into BUFR start directly after landing to enable data transfer of the complete flight to the GTS around 13:00 UTC.

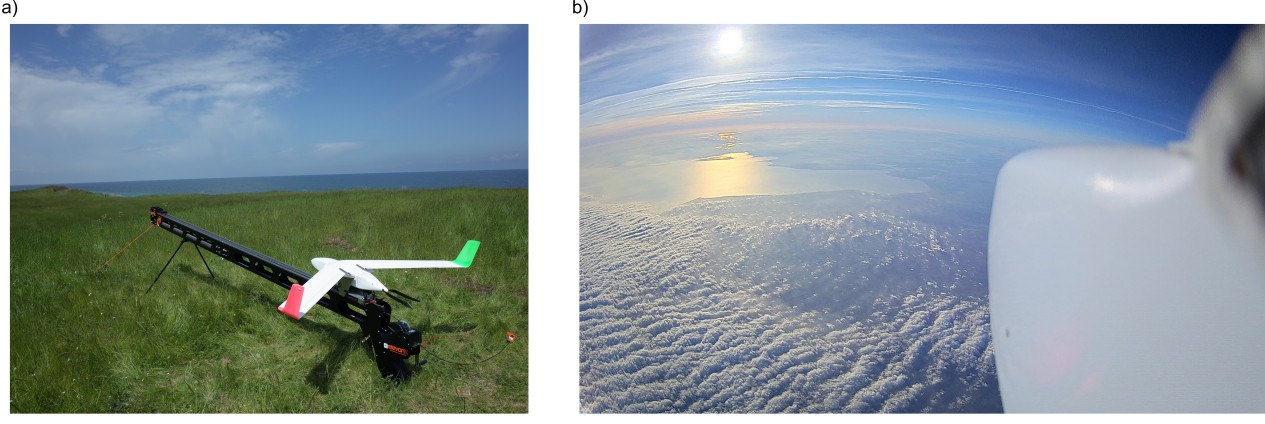

**Figure 3.** a) The drone *LUCA* mounted on the catapult to get airborne for measurements over the Baltic Sea
b) A picture out of the drone *LUCA* 10 km above the Lübecker Bight, Baltic Sea, on 28 October 2021.

and lost in low visibility conditions of drifting and blowing snow in case of high near-surface wind speed, as occurs frequently in the Antarctic. For the launch and retrieval system, no specific operation limits are in place regarding wind speed, as long as the launch and the landing is performed against the wind direction. Preliminary operating limits are therefore bound to the nominal airspeed of the drone. To address the risk of disintegration in turbulence, the airframe was constructed to resist gusts according to the EASA Certification Specifications 23.333 (EASA, 2022). On the avionics side, the systems on the drone were

245

widely predefined by national regulations (e.g., the redundancy of the command and control link). Bärfuss et al. (2022) reveals more technical details of the drone and its ground systems, which is not the scope of this journal.

## 2.3 Simplistic sensor package

An overview of the sensors and their accuracy according to the datasheet is provided in Table 1. The placement of the sensors is based on the experience with similar drone-based systems (e.g., Bärfuss et al., 2018; Lampert et al., 2020; Bärfuss et al., 2021a), and the sensor behaviour and possibilities of correcting e.g., sensor response time, are well known. The performance of sensors and the data quality is assessed directly from the atmospheric measurements, without including wind tunnel tests for the overall setup.

To enable the autopilot sensors to measure dynamic and static pressure, a total pressure port (Pitot tube) and static pressure ports on both sides of the drone were implemented in the nose section of *LUCA*. Below the pitot tube, an air inlet for the closed sensing path for the temperature/humidity probe (T/RH probe inlet) is installed, as sensor installation is known to influence the measurements (Jacob et al., 2018; Inoue and Sato, 2022). The combined resistance temperature and capacitive humidity sensor HMP110 (Vaisala, Finland), providing measurements of temperature and relative humidity in the closed sensing path, was mounted within the bulkhead, which separates the sensor chamber from the fuselage area. Subsequently, to passing by the sensing elements, the air perfuses the measurement chamber and vents out through apertures in the bottom of the nose section (Venting). This ensures well-defined pressure conditions at the sensor location close to total pressure while maintaining ventilation of the sensing element. The ventilation rate, which is assumed to be still higher than the ventilation rate of radiosonde sensors, minimises the response time of the sensors as the sensor is exposed to an increased amount of air per time compared to radiosondes. The enclosure furthermore protects the sensors from damage due to ground contact, even during rough landings.

An illustration of the sensor installation used for the measurement flights is provided in Figure 4a, where the measurement nose of the drone is shown. As the body of the combined temperature and humidity probe is made of stainless steel and is mounted through the bulkhead between the sensor chamber and the fuselage area, thermal conduction into the sensor chamber which might affect the measurements is expected in the case of temperature differences between the ambient air temperature and the temperature inside the fuselage, as mentioned in, e.g., Lampert et al. (2020). Additionally, the response times of the humidity measurements are expected to increase further, as the "wetted" area inside the sensor chamber is not well vented. Besides the sensors and pressure ports in the nose section, measurements of static and dynamic pressure, attitude angles, earth-related position and velocities – all taken by the autopilot system (Cube Orange, HexAero, Singapore) were recorded to derive atmospheric variables. As a drawback, the inertial navigation algorithm running on the autopilot system to calculate attitude angles, earth-related position and velocities, relies on industrial grade GNSS (Global Navigation Satellite System), rotation rate, acceleration and magnetic field sensors, which limits the accuracy of the location and attitude estimation and subsequently limits the accuracy of the wind calculation, varying with the flight trajectory. Particularly, heading information might be adversely deteriorated by electromagnetic interference originating from the power train, which is more pronounced during the ascent. Additionally, the estimation filter in the inertial navigation algorithm is not able to observe sensor (e.g.,

turn rate sensor) errors well during flight phases with low trajectory dynamics. Measured parameters including datasheet
uncertainties for both the drone *LUCA* and the radiosonde launched for data intercomparison are summarised in Table 1.

**Table 1.** Sensor summary for the radiosonde and the drone *LUCA* using datasheet values.

| Sensor uncertainty | Radiosonde GRAW DFM-09 | LUCA |
|---|---|---|
| position | $< 5\,\mathrm{m}$ [a] | $2.5\,\mathrm{m}$ [a] |
| pressure | $< 30\,\mathrm{Pa}$ | $< 50\,\mathrm{Pa}$ |
| temperature | $< 0.2\,°C$ | $0.4\,°C$ [b] |
| humidity | $< 4\,\%\ RH$ | $< 4\,\%\ RH$ [b],[c] |
| wind speed | $< 0.2\,\mathrm{m\,s^{-1}}$ | n/a [e] |
| wind direction | n/a [d] | n/a [e] |

[a] slightly differing between vertical and horizontal
[b] datasheet measurement range -40 °C … 80 °C
[c] including non-linearity and repeatability
[d] depending on wind speed
[e] complex error behaviour

As an example for a flight trajectory, the measurement flight on 28 October 2021 07:20 UTC, is shown in Figure 4b. The trajectory during the measurement flight consist mainly of circular patterns, and therefore provides trajectory dynamics to facilitate the calculation of aircraft attitude, velocity, and position performed by the inertial navigation algorithm.

A cut-out in the left wing is foreseen for an icing sensor as shown in Bärfuss et al. (2022), but other instruments can be
fitted into it. As the risk of in-flight icing was negligible during the measurements, a camera was installed into the cut-out. The camera captured video and audio during the measurements, see Figure 3b, and helped to analyse the behaviour of the flight controller and the motor controller of the drone.

## 2.4 Permissions for operation

The most important aspects for drone flights is safety. Operational requirements are different for each region of the world,
however, in Europe, there are now unified regulations for drone operation (EASA, 2022). The requirements concerning redundancy and the level of integrity of the drone depend on the overall risk analysis. For relatively small and light-weight drones performing flights beyond visual line of sight, which is the case for any system operating up to an altitude of 10 km, the drone falls in the category "specific", and precautions to avoid damage to a third party have to be met. An operations' handbook includes e.g., regular checks and maintenance of vital parts like motors and propellers, redundancy in the electric system,
regular training of the crew, independent control links, and many more aspects defined in the so-called Operational Safety Objectives. The flight tests were therefore done in military restricted areas, in this case close to the Baltic Sea, Germany, and the internationally recognised high quality portable radiosonde system of GRAW, Germany (Nash et al., 2011) was deployed from the launching site for direct comparison. In the restricted areas, a cooperation with the German Federal Armed Forces is

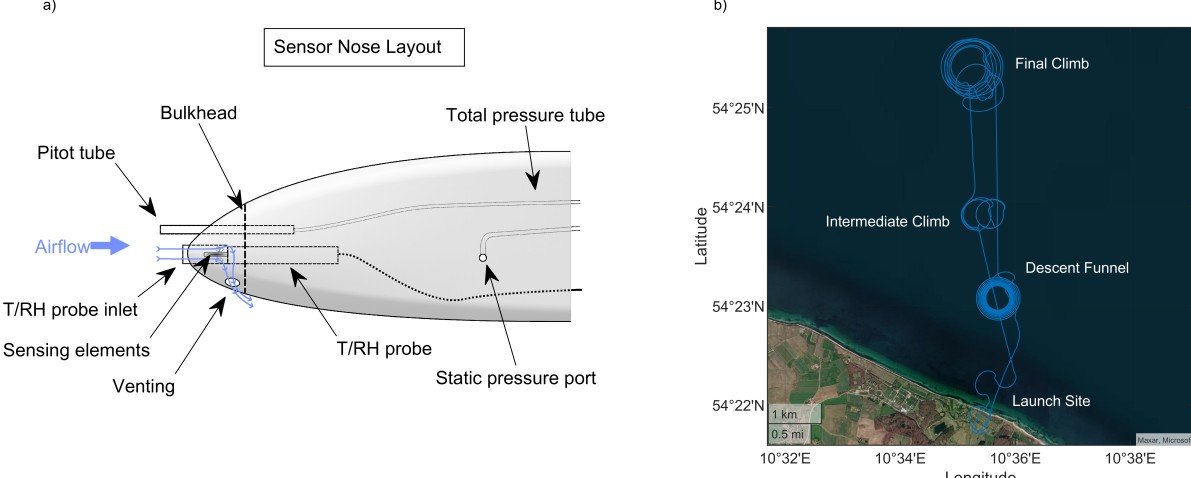

**Figure 4.** a) The measurement nose section with pressure ports for total pressure (Pitot tube) and static pressure (Static pressure port) and the installation of the temperature and humidity probe (T/RH probe, HMP110, Vaisala, Finland). The probe itself is mounted in the bulkhead, which separates the sensor chamber from the general fuselage area. The air enters the air inlet (T/RH probe inlet), passes the sensing element, and vents out through apertures in the bottom of the nose section (Venting).

b) Trajectory of the measurement flight on 28 October 2021 07:20 UTC – besides the launch site, the intermediate climb position where the drone climbs up to 3 km, the final climb position where it climbs up to ceiling, as well as the descent funnel are labelled.

required, and flight permission of the German Federal Agency for Air Traffic Control (Bundesamt für Flugsicherung, BAF) is
300 necessary.

For the operation in the Antarctic, a thorough risk assessment was performed with particular emphasis on safety and redundancy to avoid damage to third party even in such a sparsely populated area. Further, for the deployment in the Antarctic, a permission of the German Environmental Agency (Umweltbundesamt) is required, ensuring that no material stays in the pristine environment, and that the penguin population near the Neumayer Station is not disturbed.

**2.5 Method to assess the data quality of the simplistic sensor setup**

For the comparison of data obtained by *LUCA* and radiosonde data, a procedure similar to what is presented in Wagner and Petersen (2021) is applied to the data. In a first step, data within pressure bands of 2 hPa are found for drone observations and radiosonde observations. Assuming a constant wind situation at the particular altitude associated with the pressure band, the air parcel measured by the radiosonde is shifted with the wind according to the time difference between the drone measurement
and the radiosonde measurement. As for example the drone was measuring at 09:20 within the pressure level of 500 hPa and the radiosonde was measuring at 09:30 at the same pressure level, the radiosonde position is shifted backwards by the distance an air parcel would have been travelling within these 10 minutes in constant wind at this particular pressure level. This leads to a "source" position of the air parcel measured by the radiosonde at a specific time – the time when the drone was measuring

at the same pressure level (altitude). These virtual positions are then taken into account to select measurements collocated in space and time. The collocation conditions identical to the conditions used in Wagner and Petersen (2021) (spatial distance less than 50 km, temporal gap less than 30 min) were then applied, before measurements were intercompared. No explicit filter for outliers was applied, but the drone data were implicitly filtered for outliers by median filtering the data measured in each particular pressure band of 5 hPa width. Differences were computed for all pressure bands on all six flights.

## 3   Results

In the following section, the technical achievements and the resulting potential of drone measurements are presented. Therefore, the data are discussed in a qualitative and a quantitative way.

### 3.1   Technical Achievement: First flight up to 10 km with a battery powered meteorological drone

*LUCA* designates a new type of small fixed-wing drone. The combination of a relatively small wingspan below 2 m and a total weight of more than 6 kg leads to a high minimum flight speed of $18 \, \mathrm{m \, s^{-1}}$ at sea level for obtaining the lift required to fly, and therefore complicates the process of getting airborne and landing safely compared to aircraft with larger wing area related to the total mass. During the flight test and measurement campaign, the design altitude of 10 km was reached. Ascending to such altitudes can be regarded as unique for an electrically powered fixed-wing drone without solar panels. To the authors' knowledge, it is the first time that a meteorological drone powered only by electrical batteries reached the altitude of 10 km. In addition, the landing manoeuvre has been proven to be repeatable during test flights without manual control. A list of flights performed can be accessed in Bärfuss et al. (2022). During flight tests at sea level, a maximum airspeed of more than $60 \, \mathrm{m \, s^{-1}}$ has been reached, indicating that the drone is able to operate in equally high wind speed (Pinto et al., 2021). For operations, a safety margin has been applied, and the designated horizontal wind speed limit for normal operations was set to $28 \, \mathrm{m \, s^{-1}}$, which equals the nominal horizontal component of the true airspeed during the ascent. Drifting away from the measurement location is prevented by forcing the minimum ground speed to $2 \, \mathrm{m \, s^{-1}}$. The flight controller increases automatically the airspeed when the minimum ground speed falls below the threshold because of high wind speed – potentially trading the climb rate for airspeed. Resisting high wind speed and the corresponding turbulence has been demonstrated during the flight on 25 October 2021 starting at 12:12 UTC, where the maximum measured wind speed was $28 \, \mathrm{m \, s^{-1}}$ during ascent and descent at an altitude of 7800 m MSL (mean sea level). Operations in BVLOS (beyond visual line of sight) conditions and in the presence of closed cloud layers have been conducted successfully.

The drone measurements can take place from any location, up to the altitude of 10 km, if flight permission is obtained. It is completely independent of additional infrastructure, like an airport, or the availability of helium. The drone only requires a cylindrical airspace with a radius of about 11 km for the whole mission. The drone ascends and descends at a vertical speed of $10 \, \mathrm{m \, s^{-1}}$ and a horizontal speed component of $100 \, \mathrm{km \, h^{-1}}$. Therefore, the mission up to 10 km altitude and back to the landing site takes in total 33 min. The data are available by remote transfer with a temporal resolution of 1 Hz. The full data set with

345 a resolution of up to 25 Hz can be downloaded after landing and is preprocessed automatically for upload into the GTS. The subsequent launch after landing can be typically performed after a turnaround time of 20 min.

## 3.2   Qualitative comparison of meteorological observations between drone and radiosonde data

A measurement campaign with *LUCA* and radiosondes was performed at the Baltic Sea (54.4 N, 10.6 E) from 25 to 29 October 2021. An overview of the development of the meteorological conditions with altitude is shown in Figure 5 based on hourly
ERA5 reanalysis data (Hersbach et al., 2018, 2020). The distribution of relative humidity with time and altitude was highly variable during the measurement period. Also, wind speed and wind direction (indicated with wind barbs) varied with height and time. The temporal distribution of the *LUCA* flights and parallel additional radiosondes are indicated in the overview (Figure 5). *LUCA* was tested during a time period with high relative humidity, corresponding to cloudy conditions, and with high wind speed up to $50\,\mathrm{m\,s^{-1}}$, but the drone was not operated at the time of the maximum wind speed at 9 km altitude.
The overall wind direction was from West with surface wind speed around $10\,\mathrm{m\,s^{-1}}$ with absence of precipitation. Low and high scattered cloud layers were present, and a broad jet stream was forecasted just in the North of Denmark, with the jet core expected at 400 km distance to the North of the measurement campaign location.

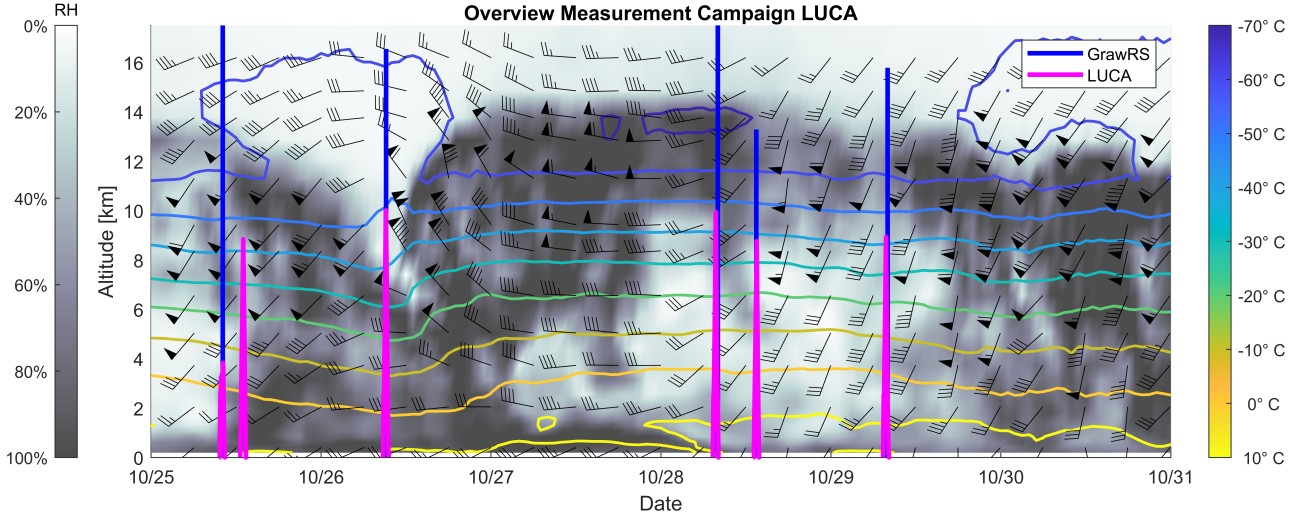

**Figure 5.** Overview of the meteorological situation between 25 and 31 October 2021 using ERA5 reanalysis data at the grid point closest to the drone measurements. The background in gray indicates relative humidity (linearly interpolated between times and levels), whereas the coloured isolines indicate air temperature. The measurement times using the drone *LUCA* are shown in magenta, the measurements with a timely and spatially collocated radiosonde (type DFM-09, GRAW, Germany) for comparison with the drone data are marked in blue. The wind barbs as defined in WMO (2011) indicate the high wind speed during the measurements, which coincide with the high variability in humidity over time.

Exemplarily, Figure 6 shows measurements of temperature, relative humidity, wind speed and wind direction recorded by *LUCA* up to 10 km. The measurements took place on 26 October 2021 at 08:45 UTC, and on 29 October 2021 at 07:22 UTC. The data of the nearly simultaneous radiosonde are shown, as well as ERA5 reanalysis data.

Each profile provided by *LUCA* shows qualitatively the same atmospheric structures as the radiosonde measurements with a similar sensor package as used on the radiosondes.

On 26 October 2021, the strong temperature inversion at 400 m as well as the transition to a different temperature gradient at an altitude of around 6500 m are equally captured by the drone and the radiosonde measurements (Figure 6a). On 29 October 2021, the temperature inversion at the top of the boundary layer at 400 m altitude is captured by the drone and the radiosonde, as well as ERA5 analyses.

Humidity was generally moderately variable during the observation period. The drone and radiosonde measurements, as well as ERA5 data, agree that there was high relative humidity (small differences between dew point temperature and temperature) up to the temperature inversion at around 400 m altitude on 26 October 2021 (Figure 6a). On both days, there were layers of lower and higher humidity. On 26 October 2021, in the lower troposphere up to an altitude of around 4 km, the individual layers of different humidity are identically resolved by the drone in altitude and magnitude (Figure 6a). On 29 October 2021, humidity measurements using the drone are in good agreement with radiosonde data up to 500 hPa (around 5 km altitude, Figure 6b). Above 500 hPa, an increasing deviation between radiosonde and drone measurements can be observed in humidity for both days. This is likely caused by the dramatically increasing response time of the humidity sensor at lower temperatures (Choi et al., 2018; Majewski, 2020) in addition to the delaying effect of the implemented closed path sensor setup. Moisture is a critical parameter to measure in the upper troposphere (Nash et al., 2011; Dupont et al., 2020).

Wind measurements agree well between the drone and the radiosonde measurements. When comparing the measurements directly, one has to take in account that the radiosonde was launched 53 min after the drone. A systematic deviation can be expected for higher altitudes due to the drifting of the radiosonde with the wind, therefore towards North-East, which resulted in a spatial distance of 40 km from the launch point when reaching the altitude of 10 km.

### 3.3 Case study: Quantitative intercomparison of the simplistic sensor setup and radiosonde data

Despite the simplistic sensor setup, data measured onboard the platform *LUCA* have shown to compare at least qualitatively with radiosonde data. To assess quantitative measures, the methods presented in Section 2.5 are applied on the data of all 6 vertical profiles (upwards and downwards) post-processed according to Appendix A. These techniques were applied for ascents as well as descents, as data during the ascent are expected to suffer from electromagnetic interference as well as other factors such as thrust line misalignment. The collocated dataset for the pressure bands of 2 hPa width from 1000 hPa to 250 hPa consists of 694 data points for the ascent and 1323 data points for the descent of the drone. Figure 7 shows histograms of the differences between the radiosonde observations and the drone observations for the variables temperature, relative humidity, specific humidity, wind speed, wind direction as well as the norm of the wind vector difference for the descending profiles. Within the histograms, average differences as well as the standard deviation of the intercompared observations are provided.

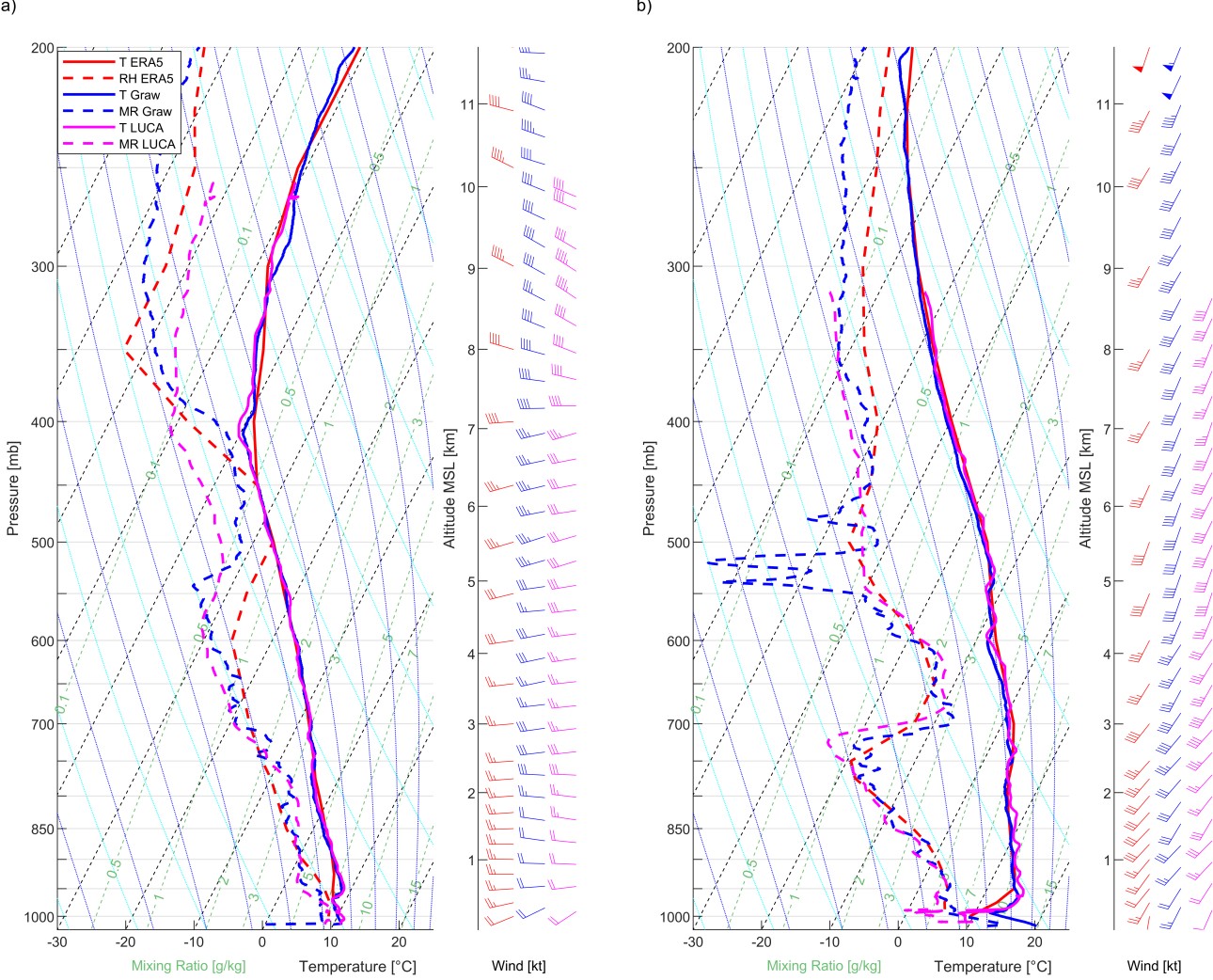

**Figure 6.** Skew-T Log-p diagrams of the vertical profiles (descent) of temperature (solid lines), dew point temperature (dashed lines) and wind (wind barbs) measured with *LUCA* in magenta, with the simultaneously launched radiosonde in blue, and ERA5 data in red. In a), the launch time of *LUCA* was 08:45 UTC on 26 October 2021, and the radiosonde was released at 09:11 UTC. The drone profile was measured during the descent from 09:09 to 09:30 UTC. In b), the drone was launched at 07:22 UTC on 29 October 2021 and the corresponding radiosonde was released at 08:01 UTC. The drone profile was measured during the descent from 07:47 to 08:12 UTC. Statistics as well as observation uncertainties are forund in Figure 8 for the profile presented in panel b).

Despite the sparse database consisting of six vertical descent profiles, the distribution of the differences for the variables temperature, relative humidity, specific humidity, wind speed and wind direction are similar to Gaussian distributions. The histogram for the norm of the wind vector difference follows a Rayleigh distribution, as it is expected for the norm of a

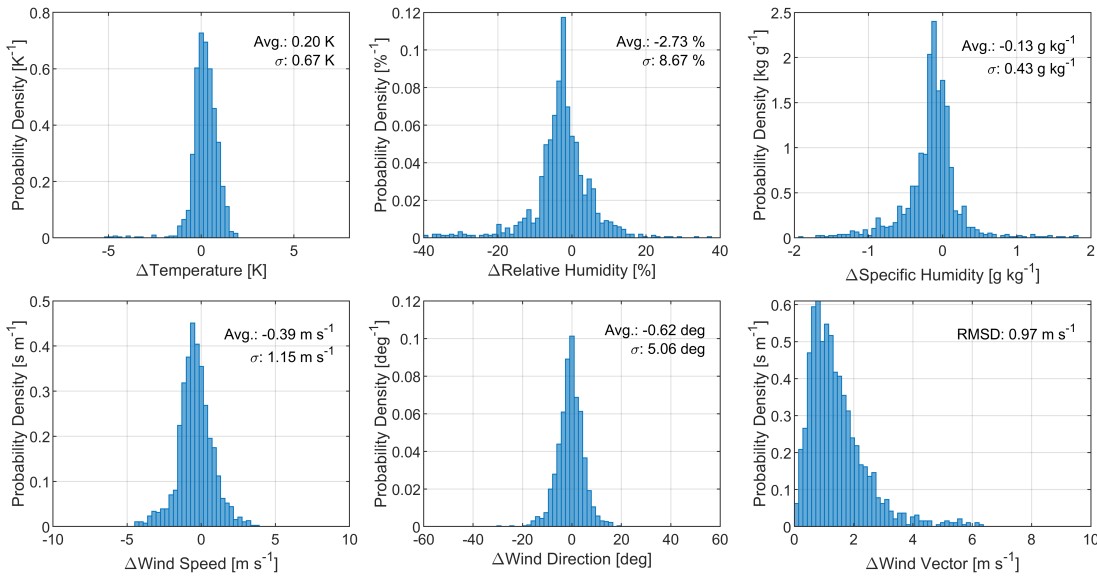

**Figure 7.** Histograms (probability density over observation difference) of the differences between collocated drone *LUCA* and radiosonde observations for various atmospheric variables. Collocation was assumed for the probed air parcels virtual spacial distance of less than 50 km and an observation time difference of less than 30 min. The virtual air parcel position was determined by virtually back-shifting the probed air parcel with the negative mean wind vector at the observation level multiplied with the time difference between the drone and the radiosonde measurements within the same pressure level band. All histograms suffer from the sparse database, but indicate a Gaussian-like distribution of observation differences, except the difference of the wind vectors, which follows a Rayleigh distribution as expected from theory. Therefore, the RMSD is shown on the histogram for the wind vector difference rather than standard deviation as for the other parameters.

two-dimensional vector, whose components are stochastically independent Gaussian processes. For the distribution of the

observation differences, the average deviation as well as the standard deviation (root-mean-squared-error for the Rayleigh distribution) were calculated and shown in the histogram plots (Figure 7). These values were retrieved for both the ascent and the descent, and are shown in Table 2.

   Comparing the statistical measures between ascent and descent data, increased differences between the radiosonde and the drone observations are revealed. This is most likely associated with a magnetic deterioration or a possibly induced side-slip

angle when the thrust line is not aligned with the drone's body (see Section 2.3 and Appendix A3). For the descent profile, the average differences as well as the standard deviation of the differences between drone observations and radiosonde observations are in the range (or even below) statistical measures shown in the study of Wagner and Petersen (2021), which includes a comparison of radiosonde data with observation data from the AMDAR/TAMDAR programme. The statistics furthermore support the values of the theoretical error estimation in Appendix C. For the humidity measurement, the average of the differ-

ences between radiosonde and *LUCA* measurements differs significantly between ascent and descent, indicating the possible

**Table 2.** Statistical measures for the differences in the collocated observation of temperature, humidity, and wind between drone measurements and radiosonde measurements. Collocation was assumed when an air parcel, virtually reverse-shifted by the mean wind and the time difference, was within a spatial distance of less than 50 km and a temporal difference of less than 30 min. The statistical measures are shown for both the descent as well the ascent profile compared to radiosonde data. During the 6 ascent profiles, an increased variation in wind observation, particularly wind direction, can be found compared to the 6 descent profiles.

| Variable | Descent | | Ascent | |
|---|---|---|---|---|
| | Average | $\sigma^{a)}$ | Average | $\sigma^{a)}$ |
| Temperature | 0.20 K | 0.67 K | 0.00 K | 0.86 K |
| Relative Humidity | 2.73 % | 8.67 % | -4.64 % | 8.16 % |
| Specific Humidity | -0.13 g kg$^{-1}$ | 0.43 g kg$^{-1}$ | -0.30 g kg$^{-1}$ | 0.53 g kg$^{-1}$ |
| Wind Speed | -0.39 m s$^{-1}$ | 1.15 m s$^{-1}$ | -0.40 m s$^{-1}$ [b] | 2.66 m s$^{-1}$ [b] |
| Wind Direction | -0.62 deg | 5.06 deg | -0.77 deg [b] | 21.88 deg [b] |
| Wind Vector | n/a | 0.97 m s$^{-1}$ [b] | n/a [b] | 2.66 m s$^{-1}$ [b] |
| Database Info | 1323 data points (68 % of dataset) | | 694 data points (35 % of dataset) | |

[a] RMSD for the Wind vector difference

[b] Wind observations during ascent are regarded as invalid due to electromagnetic interferences

need for further calibration and tuning of the post-processing parameters using a broader database in the future.

The overall uncertainties for the measured profile in Figure 6b are shown in Figure 8. As the contribution of the time lag correction and the wind calculation to the total uncertainty depends on the vertical profile itself, the total uncertainty for the
variables air temperature, relative humidity, wind speed, and wind direction is shown as a grey area around the measurements. For the air temperature $T_s$ in 8a), the radiosonde observations deviate significantly in the lower boundary layer, likely caused by the temperature sensor heated up by solar irradiance without venting on the ground before the launch. No particular uncertainty source dominates the total uncertainty for the measurements. For humidity observations in 8b), the increased uncertainty around rapid changes in humidity is visible, originating by the uncertainty caused by the uncertainty in the time lag correction. The data
sheet uncertainty dominates the total uncertainty, in which the calibration drift (up to $\pm 2\,\%$ per year according to the datasheet) is not yet included. Radiosonde observations lie partly outside the uncertainty area ($1\sigma$ plus systematic errors) for the drone measurements, potentially pointing to the need for further measurements and investigations. Wind observations in 8c) and d) compared to the radiosonde measurements are within reason regarding the uncertainty and taking into account turbulence as well as the spatio-temporal distance between the radiosonde and the drone.

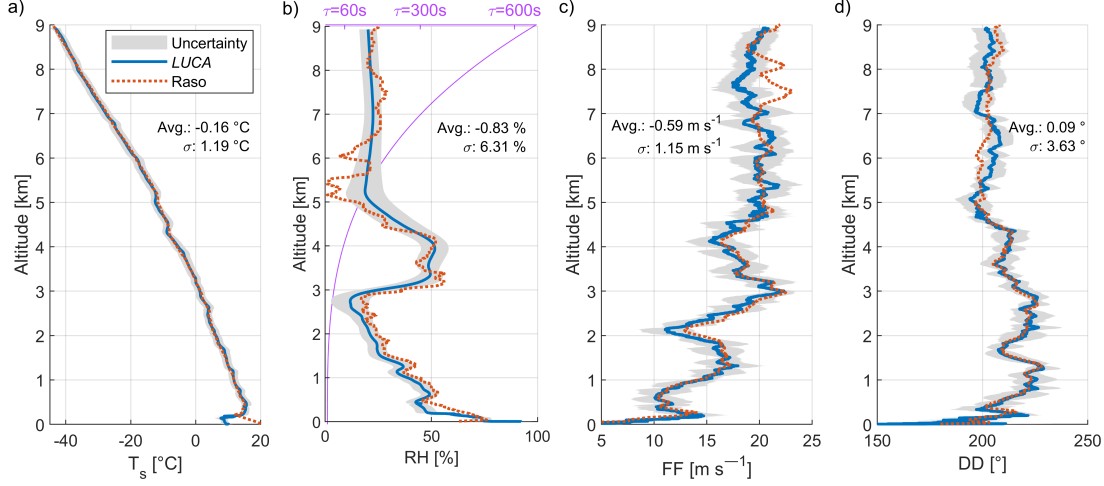

**Figure 8.** Measurements and associated uncertainties of the drone *LUCA* during the descent on the 29 October 2021 from 07:47 to 08:12 UTC and the radiosonde launched on the same day at 07:22 UTC, the same measurements as shown in Figure 6b). For the vertical profiles of static air temperature $T_s$ in Panel a), relative humidity $RH$ in Panel b), wind speed $FF$ in Panel c), and wind direction $DD$ in Panel d), the corresponding radiosonde profile is plotted. The grey area surrounding the profiles depict the estimated uncertainties of the measured variables according to Appendix C. Intercomparison statistics for the profile are added in each of the Panels. The time constant for the smoothing filter for the temperature in Panel a) is 4.2 s (roughly comparable to centred average smoothing across 120 m altitude), whereas the lilac curve in Panel b) indicates the variable "time constant" $\tau$ ($\neq$time span) for the phase neutral smoothing filter on the humidity measurements, for which no accessible replacement value for a centred average exists.

## 4 Discussion and Conclusion

The in-situ data gap in the Global Observing System has been reviewed in the introduction. Experiments on the impact of additional vertical profiles on NWP suggest that in-situ observations of the complete tropospheric column in remote areas improve forecast quality, as well as more frequent sampling of the atmosphere would do. Besides the effort in filling the observation gap in the near-surface boundary layer with drones (Leuenberger et al., 2020; Pinto et al., 2021; Inoue and Sato, 2022), the data gap with drone measurements in the free troposphere and the lower stratosphere has not been addressed – the technology of small drones has not yet been ready for observations at such high altitudes (Geerts et al., 2018; Pinto et al., 2021).

The drone measurements presented here show the potential for covering data gaps as the drone has the capability of measuring frequently, e.g., hourly, with moderate cost and effort. More frequent observations increase the forecast quality (Dabberdt et al., 2005; Faccani et al., 2009), as special atmospheric features like the diurnal cycle can be resolved. The required frequency of measurements strongly depends on the temporal variability of the atmosphere, which is highly variable for different altitude ranges, and different for each meteorological parameter, as shown in Figure D1. The atmospheric boundary layer experiences high temporal changes in temperature, wind speed and humidity. Besides these diurnal variations in the lowermost region,

temporal changes in temperature and wind speed are seen in the upper troposphere and the lower stratosphere. Processes in this layer ±5 km around the tropopause (the virtual boundary between the troposphere and the stratosphere, e.g. Gettelman et al., 2011) are known as being important for global (mass) exchange (Holton et al., 1995). Humidity, in contrast, varies highly from the ground up to the lower stratosphere with no preference on timescales, representing a chaotic system. Interestingly, temperature variability at a cycle of 6 per day is low below 5 km altitude (besides the high variability in the boundary layer), pronouncing the importance of profiling the atmosphere to higher altitudes. In NWP, physical processes are modelled to predict a future atmospheric state. The modelling of these processes performs quite well on distinct timescales to reproduce the variability of the atmosphere. The need for observations can therefore not only be defined by the atmospheric variability to avoid undersampling, but rather has to be seen complementary to the stability of the modelled physical processes and regional data gaps. Specified by data users, the WMO provides data requirements for observation spacing and uncertainty (WMO, 2015b; Leuenberger et al., 2020), which interestingly differ only little between the boundary layer and the free troposphere for both global and high resolution NWP, emphasising the benefits of profiling the atmosphere vertically with a drone up to approximately 10 km and above. Filling the in-situ observation gap both in the atmospheric boundary layer and the free troposphere will result in better forecast quality and even has the potential to further adjust and develop the modelling of the underlying physical processes.

This article presents atmospheric soundings up to 10 km altitude using an electrically powered fixed-wing drone. The developed system represents a milestone, demonstrating the capability of fixed-wing drones to perform tropospheric soundings. Measurements with a basic atmospheric sensor package were conducted. The drone measurements of temperature, humidity, and wind reported here generally agree with temporally and spatially coinciding radiosonde measurements. Minor drawbacks in the measurements occurred as expected due to the simple sensor setup.

Moisture is generally the most challenging parameter of in-flight atmospheric observations, which demands significant post-processing. This is also a known issue for standard radiosonde systems, where various corrections need to be applied as well (Dirksen et al., 2014). By design, the drone technology bears the pivotal advantage of re-using sensors and the possibility of pre- and post-flight calibration.

More advanced sensor techniques that are available could be integrated into the platform. This enables to extend the measured parameters or focus on high quality measurements of basic atmospheric parameters, e.g., by using dew point mirrors for humidity (Fujiwara et al., 2003) or fine wire temperature sensors with suitable shielding and protection (Stickney et al., 1994; Bärfuss et al., 2018). As an outlook, standard radiosonde accuracy is expected to be reached or even surpassed using a more sophisticated measurement package instead of the simplistic sensor integration used within this study. Sensor limitations and challenges are known, the needs regarding data management (Wyngaard et al., 2019) are not addressed within this article.

The system *LUCA* was designed to reach its target altitude (10 km) in conditions with average wind speed of up to $28\,\mathrm{m\,s^{-1}}$ which results in a minimum flight speed of $18\,\mathrm{m\,s^{-1}}$. The limitations concerning wind speed are related to the current maximum airspeed of $50\,\mathrm{m\,s^{-1}}$. In order to ensure the aforementioned operability, *LUCA* was constructed with suitable ground systems for take-off and landing. The mechanical start catapult enables robust starts, even in challenging conditions of high wind speed and virtually zero horizontal visibility. The landing net was designed for an automated landing manoeuvre, which

allows for landings during high surface wind and low visibility conditions.

As for now, *LUCA* does not incorporate measures to actually prevent icing, but features a dedicated sensor to detect icing. This enables safe operations of the flights, but limits the operability significantly. In future developments, this issue needs further attention.

The reported flights that were carried out demonstrate the general suitability of the technology for the envisaged purpose, explicitly covering rather challenging environmental conditions. However, systematic and extensive tests in adverse weather still need to be performed in the future. Nevertheless, the system *LUCA* was successfully operated above the design wind speed and through closed cloud layers.

Data quality assessed within a case study using a simple sensor package indicates observation accuracy comparable to AMDAR and TAMDAR observations, but further studies have to be carried out, as the observations within the case study covers only limited atmospheric conditions.

Receiving permissions for drone operations beyond visual line of sight is a demanding prerequisite for atmospheric measurement systems like *LUCA*. With its design mass of 5-6 kg *LUCA* is fairly light compared to crewed aircraft, but significantly heavier than typical radiosondes. Hence, the operational risk is an issue. By further miniaturisation of the system, both air and ground risk can be reduced, and hence is expected to simplify the process of granting permissions.

Compared to existing in-situ observing systems, the vertical profiles are similar to radiosonde ascents and descents (in some NWP centres, radiosonde descents are already assimilated, Ingleby et al., 2022) and aircraft based observations close to airports. The assimilation of drone data in the BUFR format (Ingleby and Edwards, 2015; Ingleby et al., 2022) shall therefore be quite straightforward compared to new and complex assimilation processes of, e.g., radio occultation data (Eyre, 2008). In comparison with observations from crewed aircraft, drones typically operate at a lower airspeed, which tends to result in an increased wind observation accuracy (Pätzold, 2018). Even though reaching 10 km can never replace the well established radiosonde network, the system bears the chance to augment radio soundings with more frequent drone measurements. Furthermore, in the future, it might be feasible to set up atmospheric in-situ monitoring programs by combining profiling drones as the one presented here with solar-powered semi-permanent systems in the stratosphere.

There are several possible applications and opportunities for the use of small drones measuring the complete tropospheric column and potentially the lower stratosphere – sampling the atmosphere with an increased number of observations per day with re-usable individual sensors has to be highlighted here. Re-using sensors inevitably leads to substantial knowledge about stability, strengths, and flaws of an individual sensor, and bears the chance of increasing the ability to precisely specify the uncertainty of observations, which is substantial for data assimilation. This effect is regularly observed regarding satellite sensors assessments (Rennie et al., 2021).

The use of reference radiosondes to characterise the uncertainty of NWP models to improve satellite validation and calibration is another future application (Carminati et al., 2019) to which drone measurements potentially can contribute. The quality of drone observations for sampling the complete troposphere is of superior importance, and could possibly contribute to climate applications. Regarding the involved variables, one finds that climate users tend to focus on temperature and humidity data (Ingleby et al., 2022).

For NWP applications, wind observations have arguably more than twice the impact on the quality of short range forecast compared to temperature observations (Ingleby et al., 2021). In order to augment and compete with the well established vertical profile observations (radiosondes, airliner-borne) and being used operationally, a drone must be operable by station staff of existing atmospheric observatories. Furthermore, the system needs to cope with a variety of challenging environmental conditions, including high wind speed, poor surface visibility or icing during the flight.

Upcoming implementations of such systems in the GTS will likely depend on investment cost as well as cost generated by the effort for station staff in respect to operations and maintenance. A generic approach to assess the economical benefit for drone based observations is presented in Bärfuss et al. (2022), but acquisition and operational cost are widely unclear for the system presented here, in particular regarding system failures and hardware issues.

Targeted observations were discussed controversially, as their impact strongly depends on the assimilation scheme and the NWP system (Schindler et al., 2020). Unquestionable is the use of drone measurements in contributing to scientific campaigns and the resulting reduction of cost and waste when used to replace frequent radio soundings during intense observation periods.

Although this study demonstrates the feasibility of using small drones up to the troposphere as carrier systems for atmospheric observations, envisaged extensive test campaigns (like the WMO UAS Demonstration Campaign, WMO, 2022) are needed to assess the impact of drones in forecast skills and shall increase and demonstrate the reliability of *LUCA* performing successful and safe tropospheric profiling.

*Data availability.* The data will be published at PANGAEA. Similar data sets obtained with *LUCA* up to an altitude of 4.5 km are already publicly available (Bärfuss et al., 2021a) together with radiosonde data obtained during parallel launches (Bärfuss et al., 2021b).

**Appendix A: Post-processing and calculi applied on data**

As a fundamental principle, every sensor reading a physical quantity involves a transfer function $G(s)$ from the physical quantity to the sensing element. A transfer function, utilising techniques from the field of control theory, can be noted in the complex frequency domain as

$$G(s) = \frac{Y(s)}{X(s)} \tag{A1}$$

with the input signal $X(s)$, the output signal $Y(s)$ and the complex frequency domain parameter $s$. Besides the trivial proportional transfer function $H(s) = P$ which denotes $Y(s) = P \cdot X(s)$, a common transfer function describing physical processes is the first order lag function

$$G(s) = \frac{1}{1 + \tau s} \tag{A2}$$

where $\tau$ is the so-called time constant. In the time domain, such a system has an output rate of change $\dot{y}$ proportional to the difference between the input $x$ and the output $y$, scaled with the time constant: $\dot{y} = \frac{1}{\tau}(y - x)$

An example of a sensor with an inherent first order lag transfer function is a thermistor air temperature sensor, where the heat transfer between the sensor head and the ambient air drives the rate of temperature change of the sensor head and therefore the temperature readings.

## A1  Temperature corrections

The temperature sensor head onboard the drone presented here is a thermistor element of specification Pt1000. This type of sensor naturally involves a non-uniform time lag represented by a first order lag transfer function. This implies spectral errors and input-signal dependent time lag in the temperature time series, which are in theory fully recovered using the signal reconstruction (also called inverse filtering) method. The basic idea is to apply the inverted transfer function to the measured sensor output, making advantage of a priori knowledge during the post-processing.

Expressed by a linear differential equation, the measurement process for the thermistor sensor is defined with

$$\tau \cdot \dot{y}(t) + y(t) = x(t) \tag{A3}$$

where the quantity to measure is denoted with $x$, the measurement by $y$ (both as a function of time $t$), and a time constant $\tau$. $\dot{y}$ is the derivative of $y(t)$ with respect to the time $t$. Assuming a step change on the input side and solving the differential equation, the time constant represents the sensor response time in the way that the measurement will reach 63 % of the input state within the time span $\tau$. Having identified the time constant for the sensor response, the signal reconstruction of the measurement time series $y(t)$ can be applied to determine the quantity to measure $x(t)$ by applying Equation A3. As the derivative of the measurements $\dot{y}(t)$ is needed for signal reconstruction, measurement noise will be amplified in the recovered signal, and is phase-neutrally low-pass filtered by filtering the recovered signal forward and backward (e.g., Bärfuss et al. 2018).

The spectrally corrected temperature measurements then represent the actual temperature at the stagnation point at the sensor position. In aviation, this is called "Total Air Temperature" $T_t$, and can be transferred into the static air temperature $T_s$, the temperature of the undisturbed air around the aircraft, with

$$\frac{T_t}{T_s} = 1 + \frac{k-1}{2} Ma^2 \tag{A4}$$

where $k \approx 1.4$ is the adiabatic index of air and $Ma$ the Mach number. The temperature difference between $T_t$ and $T_s$ is caused by energy conversion from kinetic to thermal energy in the air, but as the air at the sensed area with the temperature $T_{raw}$ is not subject to a complete adiabatic process, an additional recovery factor $r = (T_{raw} - T_s)/(T_t - T_s)$ in the range of about $0.6..0.95$ has to be applied to the temperature rise $T_{raw} - T_s$ when using Equation A4 (e.g., Lenschow 1972; Bange et al. 2013).

The aforementioned correction for the temperature rise is not applied to the measurements presented here, as the error is expected to be small.

An error correction rather specific to the sensor installation in the drone *LUCA* is the correction for heat transfer through the metal sensor housing between the fuselage area and the sensor chamber, as visible in the sensor package sketch in Figure 4a).

The corrected temperature $T_c$ can be assumed to behave as

$$T_c = T_r - (T_f - T_r) \cdot m \tag{A5}$$

with the recovered temperature signal $T_r$, the temperature $T_f$ measured inside the fuselage, and a dimensionless coefficient $m$ for the magnitude of the heat transfer. The coefficient $m$ which expresses the heat transfer rate from the fuselage to the sensor is regarded as independent of environmental conditions and universal for all temperature measurements with the drone *LUCA*.

## A2 Humidity corrections

Similar to resistive temperature measurements, capacitive moisture measurements suffer from a time lag which can be described by using a first order lag transfer function (Miloshevich et al., 2004; Dirksen et al., 2014; Bärfuss et al., 2018; VAISALA, 2021). In contrast to temperature sensor transfer functions, the time constant of humidity observations is heavily affected by the ambient temperature, as the diffusion process on the sensing element is slowed down in low temperatures (e.g., Miloshevich et al. 2004; Choi et al. 2018; Majewski 2020). The so-called "time constant" is therefore not constant any more, and has to be applied as a variable to the signal reconstruction of the humidity measurements. A power law was found to be appropriate to describe the "time constant" as a function of altitude (intrinsically representing temperature), and using the method of comparing ascent and descent profiles, the supporting points for the function

$$T_{humi} = a \cdot H^b + c \tag{A6}$$

which expresses the variable "time constant" $T_{humi}$ for the humidity signal as a function of altitude $H$ in meters. The humidity measurements are then corrected with the same recovery algorithm as applied to temperature measurements. For the subsequent low-pass filter, the cut-off frequency was chosen to 3 times higher than the cut-off frequency of the sensor transfer function.

## A3 Wind calculation

The basic equation to calculate the wind vector in the geodetic coordinate system noted as $\vec{w}_g = \left(w_{u_g}, w_{v_g}, w_{w_g}\right)^T$ with the wind components north, east and down defined as positive is

$$\vec{w}_g = \vec{v}_g - \vec{u}_g \tag{A7}$$

using the difference between the velocity vector of the aircraft above ground $\vec{v}_g$ and the true airspeed vector $\vec{u}_g$ of the aircraft relative to the moving air, both in the geodetic coordinate system. Although this technique is state-of-the-art since decades, see e.g. Axford (1968); Bange et al. (2013), some key information is recapped in the following, and errors subject to the simplifications made herein is described in Appendix C.

For the case that the airflow sensor is not located at the chosen reference point of the aircraft for e.g., inertial navigation where $\vec{w}_g$ is determined, the measured airflow includes induced velocities from rotational speed of the aircraft body $\vec{\Omega}_b = (p, q, r)^T$ with $p, q, r$ as rotational rates around the roll-, pitch- and yaw-axis, scaled with the lever arm vector $\vec{r}_b$ (here in body fixed

coordinates). This results in

$$\vec{w}_g = \vec{v}_g - \vec{u}_g + \vec{\Omega}_g \times \vec{r}_g \tag{A8}$$

as basic equation for wind measurements, including lever arm induced velocities. This can be prevented by calculating $\vec{v}_g$ at the airflow sensor position within the inertial navigation system or during post-processing. Equation A8 is usually deployed for research aircraft to enable high frequency wind and turbulence measurements.

The geodetic velocity vector $\vec{v}_g = (V_N, V_E, V_D)^T$ of an aircraft above ground used for wind measurements usually originates directly by the differentiation of position measurements using a GNSS (Global Navigation Satellite System) receiver

and velocity measurements of a GNSS receiver (using Doppler shift measurements for each tracked satellite). Before the deployment of GNSS, velocities were determined using an inertial navigation platform, which nowadays is optionally used to complement GNSS measurements, as GNSS data are commonly only available with a frequency of $10\,\mathrm{Hz}$.

The true airspeed vector $\boldsymbol{u}_b$ (here in aircraft body coordinates) by contrast is measured in two steps. Its measured magnitude $|\boldsymbol{u}|$ multiplied by the unit vector $e_x$ represents the true airspeed vector in the aerodynamic coordinate system $\vec{u}_a = |\vec{u}| \cdot$

$(1,0,0)^T$. Measurements of the angle of attack and the angle of sideslip with wind vanes and multi-hole probes are then used to transform the true airspeed vector from the aerodynamic coordinate system into the aircraft's body-fixed coordinate system (subscript $b$) with the rotation matrix

$$M_{ba} = \begin{pmatrix} \cos\alpha\cos\beta & -\cos\alpha\sin\beta & -\sin\alpha \\ \sin\beta & \cos\beta & 0 \\ \cos\beta\sin\alpha & -\sin\alpha\sin\beta & \cos\alpha \end{pmatrix} \tag{A9}$$

where $\alpha$ denotes the angle of attack and $\beta$ denotes the angle of sideslip. An illustration of the angles commonly used for wind

calculation is found in e.g. van den Kroonenberg et al. (2008) or any flight mechanics textbook.

Finally, applying the rotational transformation $M_{gb}$ from the aircraft body to the geodetic coordinate system

$$M_{gb} = \begin{pmatrix} \cos\Psi\cos\Theta & \cos\Psi\sin\Phi\sin\Theta - \cos\Phi\sin\Psi & \sin\Phi\sin\Psi + \cos\Phi\cos\Psi\sin\Theta \\ \cos\Theta\sin\Psi & \cos\Phi\cos\Psi + \sin\Phi\sin\Psi\sin\Theta & \cos\Phi\sin\Psi\sin\Theta - \cos\Psi\sin\Phi \\ -\sin\Theta & \cos\Theta\sin\Phi & \cos\Phi\cos\Theta \end{pmatrix} \tag{A10}$$

with the roll angle $\Phi$, pitch angle $\Theta$ and yaw angle $\psi$ enables to use the basic Equation A7 for airborne wind measurements in the form of

$$\vec{w}_g' = \vec{v}_g - M_{gb} \cdot M_{ba} \cdot \vec{u}_a \qquad \left| \vec{\Omega}_g \times \vec{r}_g = \vec{0} \right. \tag{A11}$$

neglecting lever arm effects.

Assuming $\beta = 0$, which is motivated by the directional stability of an aircraft and further assuming $\alpha = 0$ as the direction of the body x-axis can be freely adjusted to achieve this value, further simplifications follow, which are valid for either calm

conditions or ensemble observations to be averaged:

$$620 \quad \vec{w}_g'' = \vec{v}_g - |\vec{u}| \cdot \begin{pmatrix} \cos\Psi\cos\Theta \\ \cos\Theta\sin\Psi \\ -\sin\Theta \end{pmatrix} \quad \Big| \; \alpha = 0 \text{ and } \beta = 0 \tag{A12}$$

To get rid of the angle $\Theta$, which was adjusted to result in $\alpha = 0$, the mean vertical wind speed is assumed to be zero. Upon this, it follows

$$0 = V_D + |\vec{u}|\sin\Theta \tag{A13}$$

which can be used to substitute the dependency on $\Theta$ in Equation A12, leading to

$$625 \quad \vec{w}_g''' = \begin{pmatrix} V_N \\ V_E \\ 0 \end{pmatrix} - \begin{pmatrix} \cos\Psi \\ \sin\Psi \\ 0 \end{pmatrix} \sqrt{u^2 - V_D{}^2} \quad \Big| \; \alpha = 0, \beta = 0 \text{ and } w_{w_g} = 0 \tag{A14}$$

wherein

$$\sqrt{u^2 - V_D{}^2} \tag{A15}$$

can be regarded as the horizontally projected true airspeed of the aircraft.

The additional simplification using $V_D = 0$ (which can be assumed during level flight) results in

$$630 \quad \vec{w}_g'''' = \begin{pmatrix} V_N \\ V_E \\ 0 \end{pmatrix} - \begin{pmatrix} \cos\Psi \\ \sin\Psi \\ 0 \end{pmatrix} |u| \quad \Big| \; \alpha = 0, \beta = 0, w_{w_g} = 0 \text{ and } V_D = 0 \tag{A16}$$

for the north and the east wind vector components $w_{u_g}, w_{v_g}$ in the geodetic coordinate system.

The simplified Equation A16 is identical to the formulas given in the AMDAR Reference Manual, WMO (2003), Appendix I, 4., implying a potentially systematic error in AMDAR/TAMDAR measurements during ascents and descents, which is theoretically quantified during the error discussion in Appendix C.

The calculus for the true airspeed magnitude $|u|$ used for wind calculation

$$|u|^2 = R_f T_t \frac{2k}{k-1} \left[ 1 - \left( \frac{p_s}{p_t} \right)^{\frac{k-1}{k}} \right] \tag{A17}$$

as found in e.g., Lenschow (1972) is derived using Bernoulli's principle, the equation for the ideal gas and an adiabatic process. The adiabatic index of air is $k \approx 1.4$, static pressure is denoted by $p_s$ and the total pressure by $p_t = p_s + q_c$ where $q_c$ is the

640 calibrated difference between the static air pressure and total air pressure. $R_f$ is the specific gas constant for humid air, which is assumed to equal approximately the specific gas constant for dry air with $R_l \approx 287\,J\,kg^{-1}\,K^{-1}$.

Regarding air as an incompressible fluid, the calculation of the true airspeed magnitude can be simplified to

$$|u'|^2 = \frac{2\,q_c}{\rho} \qquad \left|\ \frac{1}{\rho}\left(\frac{\partial \rho}{\partial p}\right) = 0 \right. \tag{A18}$$

with the air density $\rho = p_s/(R_f T_s)$.

The drone *LUCA* measures the airspeed with a single pitot probe, leading to the simplification that the airspeed of the drone is aligned with the drone body – in other words, side-slip angle and angle of attack are assumed to be zero. Therefore, Equation A14 using $\alpha = \beta = w_{w_g} = 0$ was applied to the measurements for the calculation of the wind, and Equation A18 was applied for the calculation of the true airspeed magnitude within this study. As the measured pressure difference equals $q_{raw} = p_{t_{raw}} - p_{s_{raw}}$, the raw dynamic pressure suffers from pressure port errors, and a calibration factor $q_c/q_{raw}$ is introduced.

Besides the dynamic pressure and air density derived from the meteorological sensors, the velocity vector $\overrightarrow{v}_g$ as well as the yaw angle $\Psi$ of the drone are extracted from the autopilot system. Here, a specific filter (an extended Kalman-Bucy-Stratonovich-Filter, Kalman 1960) fuses measurements of inertial acceleration, inertial rotation, magnetic measurements of the Earth's magnetic field, GNSS position and velocity as well as pressure data into the calculation of the aircraft state.

The processed wind measurements with a data rate of 25 Hz are finally averaged over a timespan of about 5 s (data are averaged within pressure bands of 2 hPa and without excluding any wind observations e.g., during aircraft bank angles) to increase the validity of the $\alpha = \beta = 0$ assumption, as this assures to average over multiple sequences of $\alpha$ and $\beta$ oscillations.

**Appendix B:  Calibration technique**

*"While the actual sensors can be calibrated in a laboratory, the corrections needed due to flow distortion by the aircraft body require an in-flight calibration of each instrumented aircraft [. . . ]."* — Drüe et al. (2008).

This is not limited to atmospheric measurements and flow distortion. In-flight validation denotes the gold standard in measurements, as no "friendly" laboratory environment surrounds and no simplifying assumptions can be made. If any problem or unknown error source in the data is detected, a measurement system can be evaluated in a wind tunnel to identify the error source. Inversely, using simulations and wind tunnel tests can assist in designing a system, but the whole system has to be validated in-flight. As the sensors and installation approaches used in this study were derived from previously built and validated research aircraft, e.g. Bärfuss et al. (2018); Lampert et al. (2020), no wind tunnel tests were conducted initially and no unexpected error in the data during first measurements was detected justifying wind tunnel tests or even flow simulations.

Whereas manned research aircraft typically perform well described calibration and validation manoeuvres (Haering, 1990; Drüe et al., 2008; Bange et al., 2013; Cooper et al., 2014; Mallaun et al., 2015; Hartmann et al., 2018), which can also be applied to drone systems, the following techniques freely adopted from Bärfuss et al. (2018) were used within this study to determine correction parameters:

1. *Circles:* During circling at the same altitude, atmospheric variables are not expected to change, and measurements should be independent of the flight direction. This technique can also be used to estimate the error in the dynamic pressure

measurements, assuming a constant air speed of the drone and a constant mean wind field (disregarding underlying turbulence)

2. *Ascent and Descent:* The resulting profile of a variable over the altitude must be independent of the vertical observation direction and speed and match within the uncorrelated measurement error regimes. Deviations indicate the necessity to apply a time lag correction. Especially around the ceiling of the vertical profile, it can be assumed that atmospheric properties did not change significantly, and the observations obtained during ascent and descent have to agree.

    3. *Intercomparison:* Using a reference measurement system to observe the same parameters spatially and temporally close
to the measurement system to be calibrated/validated, independent measurements can be assumed, and the direct inter-comparison enables the adjustment of calibration parameters or validates the observations.

For the true airspeed calibration of the drone *LUCA*, Technique 1 using circles has been used to obtain the calibration factor $q_c/q_{raw}$, which was found to equal $0.83$ (dimensionless).

The time constant for the signal reconstruction of the temperature measurements was obtained by comparing ascents with
685 descents, Technique 2, leading to $\tau_{temp} \approx 21\,s$ for the specific temperature sensor installation. Assuming a constant lapse rate of $-0.0065\,K\,m^{-1}$ and following the fundamentals of control theory, the signal reconstruction would correct for a hypothetical hysteresis of $1.38\,K$ during the ascent. The phase neutral low pass filter (1st order Butterworth type applied forward and backwards) to reduce the noise amplified by the signal reconstruction process was set to a cut-off frequency of $0.04\,Hz$, which is roughly comparable to a central average over a vertical extent of $\approx 120\,m$.

Supporting points for the function of the time constant for the humidity sensor signal reconstruction depending on temperature in Equation A6 were approached first using the values provided in Miloshevich et al. (2004) and Dirksen et al. (2014) as well as data from recent measurement campaigns in Benin (Bärfuss et al., 2018) and Svalbard (Lampert et al., 2020). Combining the Techniques 3 and 2, significant humidity layers were identified based on intercomparison with radiosonde data, and comparing ascents with descents led to an estimate of the time constant $\tau_{humi}(T_s)$ in specific temperature realms. The function
parameters of Equation A6 then were fitted to these particular time constants in relation to static air temperature. Expressed over altitude, the supporting points for fitting the function were estimated as $\tau_{humi}= 15\,s$ at $0\,km$, $60\,s$ at $3\,km$ and $2000\,s$ at $10\,km$. The subsequent phase-neutral noise reduction filter was set to a cut-off frequency 3 times higher than the identified low-pass behaviour of the sensor installation.

To correct for the heat transfer from the fuselage to the sensing area according to Equation A5, the coefficient $m \approx 0.07$ was
700 determined by Technique 2 comparing measurements from ascents with measurements from descents.

This sums up to 6 relevant calibration and correction parameters, where the noise-reduction filters are not counted as relevant. This small number of adjustable calibration and correction parameters ensures not to over-fit observations to the expected results.

## Appendix C: Error estimation of simplifications and measurements

*Theoretical error estimation of temperature measurements*

According to Dirksen et al. (2014), systematic and random errors on radiosonde temperature observations include calibration, external radiation, convection of warm air originating from the balloon or the housing and a time lag following the physical principle presented in Appendix A. For the calibration error, datasheet values (Table 1) are applied to the drone observations with the uncertainty for the temperature sensor of $\epsilon T_{cal} = \pm 0.4\,K$ (unknown portions of random noise and absolute devia-

710 tion). External radiation is regarded as negligible since the sensor is placed within a shielded housing, and the correction for temperature-contaminated air is included in the time lag correction, as the sensor shielding and the fuselage temperature adopts to the air temperature. The prominent error source in temperature is then the uncertainty of the time constant for the time lag correction and the uncertainty in the value for the heat transfer correction from the fuselage to the sensor head.

An illustration of the effect of a time lag on a fictional vertical profile is provided in Figure C1.

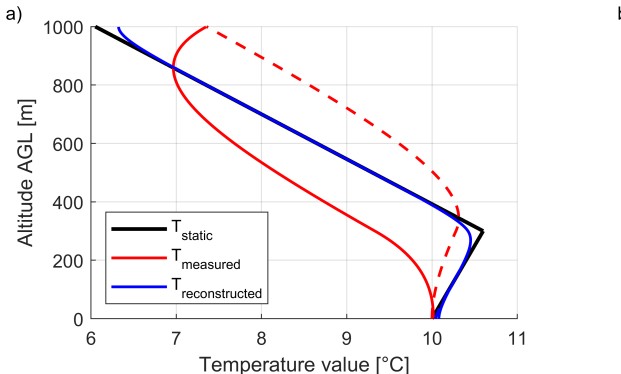 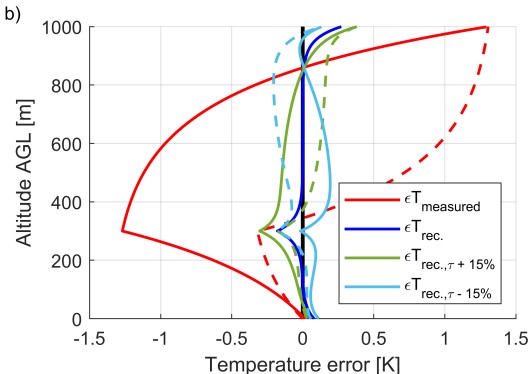

**Figure C1.** Panel a) shows a fictional vertical profile of temperature. It is assumed that the drone is moving with a vertical speed of $10\,\mathrm{m\,s^{-1}}$. In red, the theoretically measured temperature is presented for ascent (dashed line) and descent (solid line), suffering from a time lag with a time constant of 21 s. The blue profile subsequently shows the resulting profile after applying signal reconstruction and subsequently a phase neutral low pass filter. Panel b) reveals the theoretical deviation from the actual profile during ascent (dashed) and descent (solid) for the measured signal and reconstructed signals using the correct time constant $\tau_{temp}$ as well as erroneous time constants ($\tau_{temp} \pm 15,\%$)

Figure C1a shows the temperature measured with a sensor that incorporates a significant time constant and its deviation during ascent as well as descent. The minimisation of the difference between the profiles during ascent and descent then can be used to determine the time constant of the sensor according to calibration technique 2. As an additional line in blue, the reconstructed temperature profile using the inverse transfer function of the sensor behaviour is shown. At the reversal points for the temperature gradient and altitude, a small offset is observed, caused by the low pass filter applied during the signal

reconstruction process to eliminate the noise amplified by the reconstruction filter. Figure C1b then shows the correlated errors in temperature observations for the measured temperature, the reconstructed temperature, and reconstructed temperatures using an erroneous time constant for the reconstruction filter. The uncertainty in the time constant $\tau_{temp}$ is assumed to be $\pm 15\,\%$.

Following Equation 17 in Dirksen et al. (2014), the correlated uncertainty of the temperature caused by an erroneous time constant for signal reconstruction is

$$\epsilon T_{rec} = \frac{1}{2} \left| T_{meas,rec}\left(\tau + 15\%\right) - T_{meas,rec}\left(\tau - 15\%\right) \right| \tag{C1}$$

with $T_{meas,rec}(\tau)$ the measured signal reconstructed with the time constant $\tau$. The additional uncertainty caused by the smoothing filter after the signal reconstruction process $\epsilon T_{smooth}$ is roughly estimated as double the magnitude of the error introduced by the time-constant uncertainty $\epsilon T_{rec}$ for the reconstruction process based on Dirksen et al. (2014), where smoothing after the signal reconstruction using a different filter type than applied here introduces an uncertainty approximately one to two times larger than the uncertainty according to Equation C1.

The maximum error caused by disregarding compressibility (Equation A4) is determined by calculating the temperature rise for flight parameters and atmospheric values at sea level and at ceiling (10 km) and leads to a theoretical warm bias of $\epsilon T_{incompress} \approx 0.5\,K$ for the typical flight parameters.

Regarding the error incorporated by an erroneous heat transfer correction factor according to Equation A5 assumed to deviate with $\pm 15\%$, the measured temperature difference between the fuselage area and the sensor head of up to $-40\,K$ results in an error of up to $\epsilon T_{transfer} = \pm 0.45\,K$.

*Error estimation of humidity measurements*

As main error sources affecting humidity profiles, daytime solar heating, sensor lag and a temperature correction are mentioned in Dirksen et al. (2014). The error introduced by solar heating of the humicap sensor is neglected here, since an enclosure protects the sensor of solar radiation. For the calibration uncertainty, datasheet values are adopted. Regarding the time lag, the error introduced by an erroneous time constant for humidity signal reconstruction is prone to a potentially high variability and strong gradients for relative humidity in atmospheric profiles. For the estimation of the error, Equation C1 is adopted for humidity measurements and time constants, again using a deviation of $\pm 15\,\%$ from the assumed correct time constant. Besides the datasheet uncertainty of $\epsilon RH = 4\,\%$, the resulting error therefore can only be obtained using observational data.

*Error estimation of wind measurements*

Neglecting lever arm induced velocities leads to an error in wind components of magnitude $\overrightarrow{\Omega} \times \overrightarrow{r}$. For typical rotational speeds with $\overrightarrow{\sigma\Omega}$ of the drone LUCA and a neglected lever arm of $0.3\,m$, uncertainty by neglecting lever arm effects results up to an error of $\epsilon w_{w_b} = 0.05\,m\,s^{-1}$ for the vertical and $\epsilon w_{v_b} = 0.13\,m\,s^{-1}$ for the horizontal lateral (body fixed) wind component.

The assumption of zero angle of attack in the first order only influences the vertical wind speed, which will be disregarded in the following, and therefore does not introduce an additional significant error. The assumption of a zero angle of sideslip, however, incorporates an error caused by natural $\beta$ oscillations of the aircraft. For the drone LUCA, the natural oscillation of $0.7\,Hz$ with an amplitude of around $0.3°$ are irrelevant to the wind and additionally filtered out by averaging the calculated wind in a final step.

The error by neglecting the vertical wind speed, in contrast, potentially leads to significant uncorrelated errors. It is similar to the error introduced by neglecting the vertical velocity of the aircraft, as it is assumed in Equation A16. Regarding the horizontally projected true airspeed of the aircraft in Equation A14, it is evident that the relative error neglecting vertical velocity, respectively the vertical wind speed, depends on the ratio of the true airspeed and the vertical velocity / vertical wind speed. Calculating the influence of a neglected vertical speed of $5\,m\,s^{-1}$ for an airliner and a fixed-wing drone with hypothetical true airspeeds of $280\,m\,s^{-1}$ respectively $28\,m\,s^{-1}$, the deviation from the true airspeed according to the Expression A15 is $0.05\,m\,s^{-1}$ for an airliner and $0.45\,m\,s^{-1}$ for a fixed-wing drone, directly affecting the wind vector. For the error budget of the drone *LUCA*, this is neglected for normal measurements, but should be kept in mind for measurements in strong turbulence.

Another error source leading to a deviation of the retrieved airspeed from the actual airspeed is the pressure calibration coefficient $q_c/q_{raw}$. As the pressure calibration is done by an intrinsic in-flight calibration, comparing flight legs in different directions with respect to the wind, the resulting error of the calibration is below $0.2\,m\,s^{-1}$.

The error for inertial position is complex to assess, as multiple sensors are fused in an estimation filter using aircraft states at different integration levels (e.g., acceleration, velocity, and position). One can assume that the uncertainty of the velocities estimated by the inertial navigation system on the drone is less than $0.1\,m\,s^{-1}$ for each of the three velocity components in $v_g$. This is similar regarding attitude observations, of which only the yaw angle is needed for wind calculation. Attitude observations of *LUCA* fuse measurements of the Earth's magnetic field among other measurements, and therefore suffer significantly from distortions caused by the electrical system. During climb and generally when the motor of the drone is driven, no wind is calculated as magnetic interferences prevent the attitude estimation to work properly.

The uncertainty of the yaw angle observation on the drone is assumed as $5°$. From Equation A16 using the small angle approximation $\sin\Psi \approx \Psi$ yielding to $w_{v_g} \approx V_E - \Psi \cdot |u|$ for the easterly wind component, it can be concluded that the wind observation error for an attitude error equals the product of the attitude error and the true airspeed $\epsilon w_{v_g} = \epsilon\Psi \cdot |u|$. To reach a comparable uncertainty for the calculated wind, an airliner aircraft ($u \approx 280\,m\,s^{-1}$) therefore requires yaw attitude observations a magnitude lower in uncertainty than the attitude observation uncertainty of a drone with an airspeed of $28\,m\,s^{-1}$ .

Whereas uncorrelated errors (e.g., originating from the assumption of $\beta = 0$ in combination with the aircraft's eigen oscillations) are likely attenuated during the optional averaging of the calculated wind with a raw data rate of $25\,Hz$, systematic errors persist. The resulting error for the wind calculation, including both systematic errors from neglecting vertical wind and the airspeed calibration as well as the rather random error sources for inertial velocity and attitude observations, dependent on wind speed and wind direction in relation to aircraft ground course, is presented in Figure C2, which can be regarded as a graphical elaboration of the error estimation for wind measurements in Vörsmann (1984).

For both wind speed and wind direction, the error in the yaw angle is dominant. For the wind speed, the maximum error of $2.7\,m\,s^{-1}$ is seen when the air is moving lateral to the aircraft's course over ground and independent of the general wind speed, whereas the error in wind direction is maximum with the wind aligned with the aircraft's longitudinal axis (headwind or tailwind) and generally increases towards low wind speed. Neglecting knowledge about wind velocity and direction distribution, the averaged uncertainty fields of the wind observations computed for Figure C2 are $2.1\,m\,s^{-1}$ in magnitude and $7.5°$

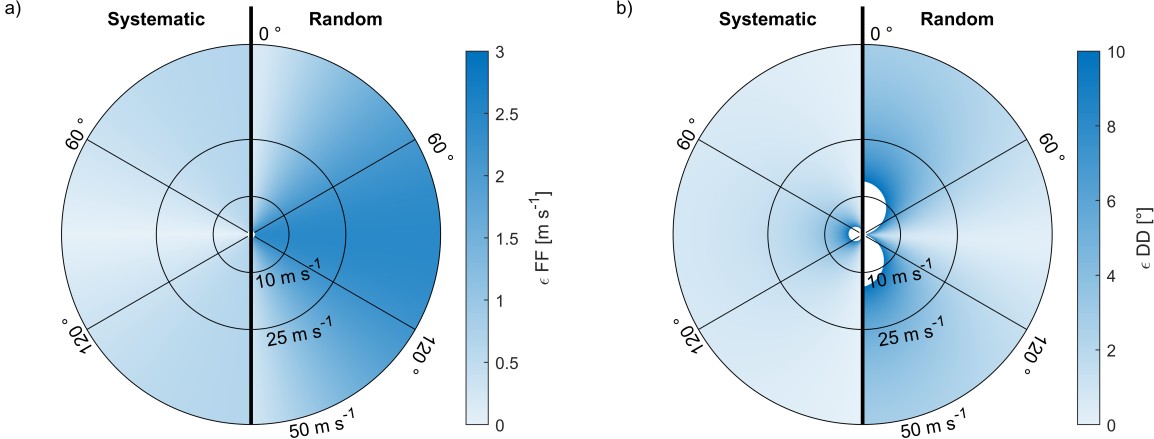

**Figure C2.** Error plots for wind observations, depending on the wind speed and wind incident angle with respect to the course over ground. The colour-coded error is further distinguished between the systematic error caused by a hypothetical vertical air movement of $5\,m\,s^{-1}$ and an error in the airspeed calibration factor (uncertainty in the airspeed of $0.2\,m\,s^{-1}$) shown in the left half of the circles, and the random error caused by inertial velocity uncertainty ($0.1\,m\,s^{-1}$ for each component) in addition with an error in the yaw angle $\Psi$ with an uncertainty of $5°$ shown in the right half. Panel a) illustrates the subsequent error for wind speed observations, whereas panel b) reveals systematic as well as random errors in wind direction observations. As the error in the wind direction reaches $180°$ for low wind conditions, uncertainties above $10°$ are whitened out in the plot.

in direction.

When it comes to peculiarities of different platform types, one can distinguish three types of platforms: aero static platforms wandering with the wind speed, Earth fixed systems such as kites or vertically ascending drones, and fixed wing aircraft of
different size and speed. Based on Equation A11, aero static platforms incorporate $\vec{u} = (0,0,u_w)^T$ with the consequence that only earth fixed velocities are needed to calculate wind. Kites or vertical ascending multicopters per definition incorporate $\vec{v} = (0,0,\mathrm{V_D})^T$, and all transformations and measurements apply to such platforms. Especially, errors in the vertical velocity assumptions might introduce substantial errors, as the horizontal true airspeed is low in low wind speed conditions for kites, tethered balloons or vertical ascending multicopters. Fixed-wing platforms, however, suffer from enhanced angular
measurement sensitivity regarding the wind calculation as shown before, which increases with the airspeed of this platform type.

## Appendix D: Variability of the atmosphere

In the following, the temporal variability of temperature, wind speed and relative humidity up to 35 km is illustrated based on ERA5 reanalyses to enable a discussion of the use of additional data observed with drones. To describe the variability of the atmosphere at a specific location and altitude for one parameter, time series of the atmospheric variables can be transformed from the time domain into the frequency domain, e.g., by applying a Fourier transformation. Mapping the results in the frequency domain for every altitude on colours and stacking them over the corresponding altitudes reveals the height dependent variance of an atmospheric variable on different time scales (herein, the timescale is denoted by cycles per day). E.g., a bright spot for the variable temperature at 1 cycle per day (period 24 hr) close to the surface reveals the diurnal cycle of the temperature in the boundary layer (forced by the sun). Such an analysis of temperature, humidity, and wind speed is shown in Figure D1 for the location of the Lindenberg Meteorological Observatory of the German Weather Service using hourly ERA5 reanalysis data (Hersbach et al., 2018, 2020) for a timespan of one year. Similar calculations are found in Vinnichenko (1970) and Fiedler and Panofsky (1970), but in contrast to the study of Vinnichenko (1970), which focuses on the kinetic energy spectrum in the free troposphere and the study of Fiedler and Panofsky (1970), which focuses on spectral gaps, the qualitative fields presented here are used to discuss the benefit of sampling the atmosphere frequently (more than twice a day).

In Fig. D1a), one can clearly see the diurnal cycle of temperature close to the surface and in the stratosphere through the bright lines at one cycle per day. The bright lines at two cycles per day shall not confuse the reader, as the analysis relies on decomposing the time signal in sinusoids with differing frequencies (cycles per day), and higher harmonics (natural products of the fundamental frequency) reveal the non-sinusoidal waveform of the diurnal cycle. Besides in the atmospheric boundary layer, increased variability can be seen around 10 km altitude for temperature, and, after a dip in activity, around 20 km. Interestingly, temperature variability at a cycle of 6 per day is low below 5 km altitude, pronouncing the importance of profiling the atmosphere to higher altitudes.

*Author contributions.* A.L. and H.S. developed the project idea and acquired funding. H.S. performed the analysis of requirements. K.B. developed the system *LUCA* and the instrumentation. K.B. performed the data processing and data analysis. All authors contributed to and reviewed the manuscript.

*Competing interests.* The authors declare no competing interests.

*Acknowledgements.* The project AEROMET_UAV was funded by the Modernity Fund (mFUND) of the Federal Ministry for Digital and Transport (BMDV) under grant 19F2072A. The results as presented within Figures D1, 5 and 6 contain modified Copernicus Climate Change Service information 2022 (ERA5 reanalysis data for temperature, wind, and moisture). Neither the European Commission nor ECMWF is responsible for any use that may be made of the Copernicus information or data it contains. The authors would like to thank

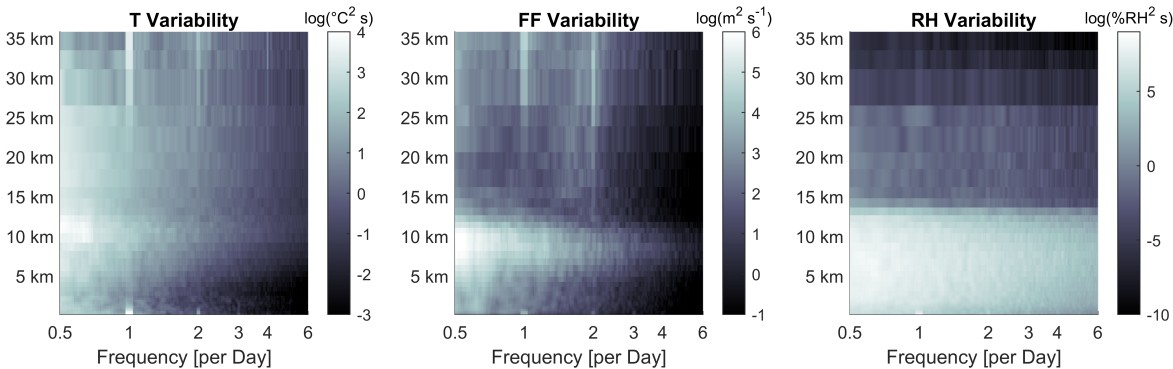

**Figure D1.** Colour coded mesoscale (periods from 1 to 48 hr (Fiedler and Panofsky, 1970)) variance densities of time series of atmospheric variables at specific pressure levels (heights). The variances are representative for the atmospheric variability dependent on height and cycles per day (1 cycle per day corresponds to a period of 24 hr, 2 cycles per day to 12 hr). The data for this analysis origin from the ERA5 reanalysis product. Bright colours indicate high variance density and hence high variability. a) represents temperature, b) represents wind speed, and c) represents humidity.

Harald Schüßler (more than scale composite); Ruud Dirksen (DWD); David Burzynski, Stephan Bansmer and Juan Velandia (Coldsense Technologies); Rolf Zimmermann and Markus Landeck (Ing.-Büros); Hans-Jürgen Unverferth (Zanonia-Flyers); Valentin Möller and Alex Helbing (Exabotix); Samira Terzenbach (AWI); and the colleagues and students at TU Braunschweig who supported the project, Heiko Wickboldt, Lutz Bretschneider, Magnus Asmussen, Andreas Schlerf, Falk Pätzold, Thomas Rausch, Sven Bollmann, Tom Schwarting and Maximilian von Unwerth. The authors would also like to thank the team of the military training ground Putlos/Todendorf under the direction of Hptm Johanssen, in particular Hptm Lucht, Kurylak and Link, for their kind assistance during the test flights.

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
