# Peer review of "Drone-based meteorological observations up to the tropopause – a concept study"

_Atmospheric Measurement Techniques, 2022_

## Author Response (AR1)

**Final Author Reply to the Editor    -        amt-2022-236**

Drone-based meteorological observations up to the tropopause
Konrad Benedikt Bärfuss et al., Atmos. Meas. Tech. Discuss.,
https://doi.org/10.5194/amt-2022-236-RC1, 2022

Dear Editor,

In the following, you will find a list of the referee comments and all associated changes within the manuscript. The differences compared to the manuscript version submitted before are highlighted within the document "LATEXDIFF_Drone_based_meteorological_observations_up_to_the_tropopause". Please note, that line numbers of changes based on reviewer comments are associated to the aforementioned document.

You will find two major changes: Within "Methods", our techniques to calculate and correct observations are described, and a quantitative assessment of the simplistic sensor setup is shown in "Results" revealing accuracy comparable to AMDAR/TAMDAR.

We hope that the quite brief notes on what changes were initiated by which of the reviewer's comments are in your interest.

Now let's start:

**RC1**

The manuscript "Drone-based meteorological observations up to the tropopause" by Konrad Bärfuss, Holger Schmithüsen, and Astrid Lampert, presents the development and first results of an uncrewed aircraft system (UAS)capable of sounding the atmosphere up to 10 km. This is an outstanding achievement and can have a major impact on improving in-situ atmospheric measurements for a future use in the operational network of a met service. The content gives a valuable contribution to the community, however some major changes have to be implemented before publication. I am going to explain why below:

Thank you for your appreciating general comment.

- no changes within the manuscript based on this comment -

General comment:
The manuscripts present the use and first deployment of a novel UAS for atmospheric sounding up to 10 km. The airborne platform is introduced and discussed, including the design envelope of the aircraft.

This is the main goal of the article, and we now emphasize more this purpose.

Addressed in 515ff and 610ff

Measuring up to 10 km is challenging as the sensors have to handle and withstand a wide range of environmental conditions. The authors explained this in much detail in Section 2.2 in their manuscript. The sensor package is mentioned, but a proper sensor introduction, or the methodology of wind

measurement including an adequate error discussion, is missing. However, due to the harsh conditions the sensors are exposed to, a detailed discussion is essential. The authors give some information in the caption of Table 1. Still, it remains unclear what is the temporal, thus the spatial or vertical resolution of the sensors or the absolute uncertainty depending on the flight trajectory or how well the sensors perform in e.g. very low temperatures, high humid conditions etc. Also, the methodology of the wind estimation remains unclear. Although the author claims that calibration and removal of installation errors are an own branch of science, validating the LUCA system with commonly used systems such as a met tower or a validation in a climate chamber is required. The authors show a comparison with radiosondes, but due to a time differences of several hours between the radiosonde and the aircraft measurement, the comparison remains insufficient and shows large deviations.

We agree with you, that presenting a validated measurement system would need a detailed introduction into sensors, sensor placement, data processing and validation techniques. Within the presented work, the ability of carrying sensors up to the tropopause on board small UAS is the key content. Besides, we show low-quality measurements to demonstrate what the system is intended for. Similar sensor packages have been deployed on other UAS and validated with well-established methods in the atmospheric boundary layer.

Based on your comment, we added more information on the simplistic sensor setup as well as the postprocessing algorithms including wind vector retrieval and humidity sensor handling. Furthermore, we included a new subsection within "Results" and intercompare data (LUCA vs radiosonde) as a case study "deploying a simple sensor setup" in the similar way data quality is assessed in Wagner and Petersen, 2021 [1].

However, the focus of the article is to present the platform as a re-usable carrier for measurements in the entire troposphere.

Sensor installation is addressed in Subsections 2.3 (esp. Fig. 4a), methodology of wind measurements in Subsection 2.5, data quality assessment technique in 2.6 and the results of the quantitative assessment is presented in Subsection 3.3.

Specific comments:
4: What do you mean by environment friendly additional data?

Compared to the environmental impact of radiosondes (waste not collected, but left in the environment) as well as aircraft powered by fossil energy (increasing the atmospheric carbon dioxide), the data is gathered in an environmentally friendly way using the UAS LUCA, that is without emitting greenhouse gases and without leaving waste in the environment.

- no changes within the manuscript based on this comment -

84: Usually UAS stands for uncrewed aircraft system

Indeed, we altered the abbreviation by mistake.

We now use the term "drone" throughout the manuscript.

102: Authors could add this citation:
Jensen, Anders A., et al. "Assimilation of a coordinated fleet of uncrewed aircraft system observations in complex terrain: Observing System Experiments."
Monthly Weather Review 150.10 (2022): 2737-2763.

Thank you for the suggestion, we implemented the reference in the revised version.

Included in 106

165: Does this also account for take-off and landing?

Yes, there is no need for a lower limit than 28 m/s during take-off and landing. We clarified this, and stated that operators have to be aware of wind gusts which could affect the UAS during these mission phases. Crucial for operations under such conditions is the automatic alignment with the main wind direction, based on the real-time measurement data.

Addressed in 235ff

203: Is 13:00 UTC or local time. Does LUCA have a real-time downlink and can provide data during the flight at 12 UTC? This is not clear here.

All times in the manuscript are provided in UTC, which is local time minus 2 h in the presented cases, we added "UTC" in the text.
Indeed this depends on the data link and possibly reserved frequency bands. The system is able to link down real-time data, but as temperature and humidity measurements are quality checked and processed using post-priori information, the data is not available in real time up to now. However, the wind speed and wind direction is available at a frequency of up to 100 Hz, which is crucial for surveying the mission. In case of high wind speed exceeding the aircraft limitations, the mission is abandoned. The wind direction is of high importance for the landing phase, as the aircraft trajectory is oriented against the wind direction for landing.

Clarified in 220 and a note on real-time data in 215/216

Section 2.2: A clearer description of the aircraft system is missing. What kind of engine, autopilot, C&C link etc. is used ?

Some more information on technical details is provided. For a more detailed overview of the technical questions regarding the system, the reader is referred to Bärfuss et al., 2022 [2], as the technical implementation is not the focus of this study.

We refer now to the technical paper in e.g. 240 and added the autopilot SW in 231.

Section 2.2: Why does the aircraft not perform a normal landing with a flare? Common open-source autopilots such as the one used in LUCA can handle this.

These technical questions regarding the system are now addressed in Bärfuss et al., 2022 [2] and are not in focus for this study. However, we added a short note that in case of high near-surface wind speed, as occurs frequently in the Antarctic, the system would be blown away and cannot be found in low visibility conditions of drifting and blowing snow. Besides the harsh surface in the Antarctic, a soft belly landing with a flare would require a reliable altimeter, working in conditions such as snow drift, and would increase system weight and cost.

Clarified in 232ff

Section 2.3 How is the measured data stored?

The data is stored directly in the autopilot system.

- no changes within the manuscript based on this comment -

218: 'closed sensing path'. An illustration or a picture of the sensor system would be very helpful. What is the mass flow around the sensor?

In the revised version we provide an illustration. The flow speed around the sensor is around 5 m/s (see Bärfuss et. al., 2018 [3]).

Illustrated in Figure 4a, additional information on the sensor system in Subsection 2.3

219: For what errors are they corrected?

The data is corrected for dynamic sensor response – in the case of the humidity sensor including a variable "time constant" of the assumed (simplified) underlying transfer function G(s)=1/(1+Ts). The basic method using a constant "time constant" is explained in detail in Bärfuss et al., 2018 [3], which is now referred to in the revised manuscript.

We decided to insert a dedicated Section: "2.5 Postprocessing and calculi applied on data"

221: How does the flight trajectory of a mission look like? The authors should show the flight trajectory at least of one flight mission.

A good idea! We now included the flight trajectory of a mission.

See Figure 4b

221: What do you mean by heading output? How is the horizontal wind, using the heading information, calculated? Please explain the method of your wind estimation in detail and please elaborate on:
- How does the flight trajectory impact the wind estimation, and how does this effect the measurement error?
- How does the wind speed influence the wind speed uncertainty?
- Is the wind measured during turns or during the ascent or descent in straight flight?
- During which trajectories is the wind estimation bad?
- What is the overall error of the wind estimation, and how do you calculate it?

The calculation of the wind speed is now mentioned briefly:
The wind is calculated similar to the wind estimation of AMDAR/TAMDAR – as a difference between the inertial trajectory vector and the wind vector. Similar simplifications as for (T)AMDAR are applied and consist of:
- Zero angle of sideslip
- Zero vertical wind
In the AMDAR/TAMDAR wind retrieval, measurements during bank of more than 5 degrees are excluded. For UAS flying at much lower air speed (one order of magnitude lower airspeed), the sensitivity of angular errors are no that crucial, as all three vectors (wind, airspeed, ground speed) are on the same order of magnitude (note that airliners operate at higher airspeed and therefore the vector difference of two large vectors is more sensitive to angular errors).
The flight trajectory has an impact mainly on two measurements: Heading (as calculated within the inertial navigation system) and sideslip angle. Both have an impact on the resulting vector difference and are heavily system-dependent. Elaborating the quantity of the error would require system identification of the aircraft and INS-filter simulations – both are regarded as an inadequate effort with respect to the main statement of the article ("small UAS are capable of sounding the complete troposphere"). Qualitative considerations instead should be implemented along with a description of the wind calculation.
In stationary flight just after dynamics (to feed the INS algorithm to compute a useful heading - on straight legs, the heading usually is tied more or less slowly towards the vector over ground), the wind estimation is expected to perform best. Spiralling might be regarded as a quasi-stationary flight state, but still provides dynamic input to the INS filter. In increasing wind speed, the spiral is deformed into a short-time curved and a long-time straight trajectory. This might lead to reduced absolute quality of the wind vector components in high wind speed, but stable relative quality.

Up to now, we do not calculate the wind error in flight and rather estimate/observe the error during intercomparison/assimilation. We included the method for the wind estimation in the manuscript and added an assessment of the wind error within the section "Results".

This rather complex comment is addressed by the Subsection 2.5, 2.6 and 3.3 "Case Study: Quantitative intercomparison of the simplistic sensor setup and radiosonde data"

224: In addition, magnetic vector measurements to be fused in the attitude estimation might be deteriorated. Is the magnetic sensor fused or not?

The magnetic sensor is fused – but as you mention, one has to be careful using it. While the distortion can be eliminated to some extent, we rather use information from the magnetic sensor during the descent, where the generated electromagnetic fields through the power train (including cables) are minimal.

Also addressed within 268ff, the effect of magnetic deterioration is visible in the data quality assessment in Table 2 (Descent vs. Ascent)

225: How can a camera be a reliable ice detector? Please explain in more detail.

The phrase is written misleading. The camera is not an ice detector, but a physical replacement for the cut out in the wing, which is usually used for the ice detector. The cut out is foreseen for the sensor, but other instruments can be fitted into it.

Rewritten and clarified, 275ff

Table 1: A real discussion of the sensor error in the text is missing. The authors give some information in the caption of Table 1, however this is hard to follow and mainly cites further publications using similar sensor but on different platforms. It remains unclear what is the temporal, thus the spatial or vertical resolution of the sensors during the ascent and descent, the absolute uncertainty depending on the flight trajectory and the relative error (see also my comment for line 221). Although the author claim that calibration and removal of installation errors are an own branch of science, at least a comparison of the LUCA system with commonly used systems such as a met tower should be performed to enable a real sensor error estimation or a validation in a climate chamber. Also, it is not clear what is meant by calibration and removal of installation errors.

The focus of the article is on the development of the platform and the technical achievement, and not a detailed discussion of the sensor error. Nevertheless, we included an assessment of the data quality as part of a case study "Assessment of data quality using a simplistic sensor setup on the platform LUCA" within the results section.

Included and discussed within the quantitative data quality assessment (Subsection 3.3) as well mentioned in a short note in the discussion (572ff)

257: '... which equals the minimum horizontal component of the true airspeed during the ascent.' Why minimum?

As the dynamic pressure to stay airborne limits the true airspeed at higher altitudes (due to the decreasing air density), the horizontal component of the true airspeed is higher at the ceiling than close to the ground. Therefore we used the wording "minimum" to state that we never fly at a lower horizontal airspeed component during the mission – which subsequently gives us the wind resistance up to this true airspeed.

Replaced by "nominal" for clarification, line 397

265 ff: Part of this has been mentioned also earlier in the text. I would recommend skipping this and refer at an earlier stage to the aerospace journal, e.g., in Section 2.2.

Thank you for recommending so – we agree that we should clarify this point at an earlier stage ad should not double information at this point.

We removed the doubling sentences, and refer to the technical paper earliest in Subsection 2.2

261: Where did the flight take place? An overview of all analysed flights described in this study should be presented.

That is a good hint; we implemented more information on the flights.

Instead of doubling information on all flights, we now refer to the technical paper [2] in line 394

283: The word simultaneous is misleading, as even the authors claim that there is a time gap up to three hours between the flights and the radio sonde ascent.

Thank you for the hint. We will make reference to a study, where TAMDAR data was compared to radiosoundings and apply the metrics therein to determine whether the spatiotemporal gap is low enough to compare data [1]. The flight with a time difference of three hours is not used for intercomparison.

We removed the first flight during the campaign (25.10.2021 09:21 UTC) and the flight without quasi-collocated radiosonde measurements (25.10.2021 10:07 UTC) from Figure 7 but added another profile (29.10.2021 07:22 UTC).

294 and 301: Plots in Fig. 5 are too small to identify the ABL structure and follow the explanation in this paragraph. I would suggest increasing the resolution at least for the lower part of the atmosphere.

As the study does not focus on the ABL in contrast to most previous UAS measurements, we choose the standard representation for radiosonde data – although the ABL structure cannot be identified by this plot type.

We increased the resolution by removing a panel and stretching the Figure height.

305: 'multiple times' What is the frequency? See also comment for Table 1.

Thank you for the hint – we implemented information about how turbulence can be represented independent of the measurement frequency (Eddy Dissipation Rate) and added the current measurement frequency.

The measurement frequency is now mentioned in 414/415. Finally, we decided to not include a turbulence parameter within the manuscript, as we do not see the benefit of adding another subsection.

Fig 6. Part of the caption belong to the Section. It is not clear what is meant by 'on decomposing the time signal in sinusoids with differing frequencies (cycles per day), and higher harmonics (natural products of the fundamental frequency) reveal the non-sinusoidal waveform of the diurnal cycle' A more detailed explanation should be already implemented in the Method Section of the study. Also, it remains somehow unclear to me, what the benefit of Section 3.3 in relation to LUCA is. For instance, the sentence 'Interestingly, temperature variability at a cycle of 6 per day is low below 5 km altitude, pronouncing the importance of profiling the atmosphere to higher altitudes', which only appears in the caption of Fig. 6 should be stressed more in the text.

The benefit of the section is the representation of the intensity of changes within the atmosphere on diurnal timescales. We added more information about the decomposition/transformation of the atmospheric variables from time- into frequency-domain and moved the subsection into discussion/outlook.

The Subsection has now partly moved into "Methods" (Subsection 2.7) and is referred to within the Discussion (516ff)

350: Minor drawbacks in the measurements occurred as expected due to the simple sensor setup. What are these? This is not clearly described in the previous Sections.

We included more information on the drawbacks such as heat transfer from the fuselage.

See Subsection 2.3, 254ff

354ff: 'Using a more sophisticated measurement package, standard radiosonde accuracy is expected to be reached or even surpassed. By design, the UAS technology bears the pivotal advantage of re-using sensors and the possibility of pre- and post-flight calibration....' This is very speculative and should be addressed as an outlook.

We agree with the author, and put the statement into the outlook.

Now declared as outlook, line 556/557

367: I disagree with this to claim a camera a 'dedicated sensor' (see also comment above)

The sentence is indeed misleading, so we changed it.

As we clarified this before (275ff), we did not changed the sentence in line 565

Technical comments:
2: In the ABL, above the oceans and in polar regions
146: ... the UAS of type LUCA

We implemented these comments in the revised version.

Implemented in line 2 and 153

Thank you for the attentive reading and your valuable comments which enabled us to further improve the manuscript!

**RC2**

The Authors would like to thank the referee for taking the time to review our work – which is highly appreciated in such busy times.

Preface:
Despite the valuable comments and suggestions, we had to discern that the focus of our study might have been misinterpreted here - possibly caused by the title and the lack of clear statements regarding the focus. We now state clearer, that the aim of the study is to provide an introduction to Aircraft Based Observations with small UAS and to present first meteorological observations of a self-powered small UAS using a simplistic sensor setup to demonstrate today's capabilities of small UAS as a "carrier platform".

Let me begin by saying that I am a strong supporter of the utility and benefit of automated aircraft reports to fill the time and space gaps left between other in situ observing systems. I therefore read the article entitled "Drone-based meteorological observations up to the tropopause" by Bärfuss, Schmithüsen and Lampert with interest. Unfortunately, I came away from it feeling disappointed and uninformed. As it stands, the text reads much more like a "Concept Study" or perhaps "A Report on Preliminary Project Activities" than a scientifically supportable journal article and needs to be substantially revised and enhanced before it can be considered for publication. Some of my reasoning for this decision is summarized below.

Firstly, the authors have dedicated nearly 1/3 of the article to the Introduction and background materials. And even then, they have not included any discussion of problems that other authors have discovered with automated reports from commercial aircraft that should provide a far more stable platform for collecting data and providing representative measurements or the atmosphere without being affected by artifacts related to aircraft stability. For example, it has been documented that AMDAR observations from longer range aircraft provide more accurate wind reports than TAMDAR reports obtained from smaller/lighter regional jets, whose wind reports are much more susceptible to the effects of turbulence than the larger/heavier aircraft. See recent papers by Wagner et al. on this topic. The excessive presentation of background material also detracts from the limited analysis of the flight results.

You might refer to Moninger et al. 2010 [4], which states: "The lower quality of the wind data from TAMDAR is likely due to the less accurate heading information provided to TAMDAR by the Saab-340b avionics system. Accurate heading information is required for the wind calculation, and the Saab heading sensor is magnetic and known to be less accurate than the heading sensors commonly used on large jets."
Literature in general links accuracy to the avionics/navigation systems used to calculate wind in AMDAR/TAMDAR – the accuracy is not linked to the size of the aircraft, which is also reflected by several small research aircraft flying at low altitude, which provide the wind vector with high quality (e.g. D-IBUF / D-ILAB / HALO D-ADLR / Polar 5 C-GAWI / Falcon D-CMET etc.)

Using the simple (T)AMDAR approach to estimate wind (pure vector difference, see [5]), no rotational effects are included, and observations are taken as invalid above a certain aircraft roll threshold. In addition, the side slip angle and the angle of attack is assumed to be constantly zero.

Doing so, turbulence definitely folds into measurements, but only because of the algorithm and simplifications applied – depending on e.g. the averaging time span and aircraft specific motion dynamics. A discussion of such peculiarities depending on the algorithm would be outside the scope of the article. Nevertheless, we included more details about how wind measurements were derived and finally decided to include a more detailed data analysis of the measurements as a case study using a simplistic sensor setup on a small UAS.

The Introduction is now significantly less than 1/3 of the article.
Clarification on wind measurements is now added information about the wind calculus (Subsubsection 2.5.3, line 232ff), referred to AMDAR/TAMDAR (344ff) and concluded with a thought how wind measurements are affected by the airspeed and angular errors of the measurement platform (351ff). The mentioned papers were referred to as we now use a similar approach to intercompare drone data and radiosonde data.

A fundamental issue of terminology is also scattered throughout the paper related to the phrase "Uncrewed Aerial Systems" (or UAS). According to online references on drones, "a UAS or "Unmanned Aircraft Systems" includes not only the UAV or Drone but also the person on the ground controlling the flight and the system in place that connects both. Basically, the UAV is a component of the UAS, since it refers to only the vehicle itself." As such, why is yet another (possibly conflicting) terminology being added the already excessive meteorological acronym list?

We are well aware of the discussion on terminology. Around 10 years ago, the term unmanned aerial vehicle (UAV) was the standard term to describe the unmanned aircraft, while the term unmanned aerial system (UAS) referred to the whole system, including in particular the ground-control station. In those times, the term drone was mostly used for military unmanned systems. To overcome the different terms, the term remotely piloted aircraft system (RPAS) got more familiar. For gender reasons, the term UAS was spelled out "Uncrewed aerial systems" (one might refer to the acronym "uncrewed spacecraft") to keep the well-known acronym. Today there is a tendency of using the term drone to avoid gender conflicts. For simplicity, we now use the expression "drone" throughout the manuscript.

As mentioned in our reply, we now use the term "drone" throughout the manuscript.

When compared to other sections of the paper, the introduction length (143 lines) seems to be excessive. For example, the important information about instrument accuracy in the section titled Sensor Package uses only 13 lines (and includes a Table that is never referenced in the text and contains an undefined term "prospective"). For other aircraft observing systems, detailed air tunnel tests were performed with the full system to determine whether the installation approaches would degrade the observations further. The reader can only assume that no such tests were done of the LUCA installation approach. The only vague reference notes that some temperature observations were affected by "heat transfer inside the fuselage to the sensor head". The authors also assume that the HMP110 sensor package will work as well on a drone moving through the atmosphere (at a speed not clearly specified in the paper) as it does on a radiosonde package suspended from a balloon moving with the atmospheric flow, not through it. This should not be considered a given.

As not everyone interested in the article is aware of the data base for numerical simulations, we disagree on the introduction length. The introduction is of high importance to motivate the development of our system, as there is currently no such tool available to perform this kind of measurements.

We totally agree that sensor selection and installation are very important. We now point out in the manuscript more clearly that we have used and characterized the same types of sensors on drones flying at similar air speed, and methods how to take care of limited sensor response time. However, the purpose of the manuscript is to present the novel development of the drone covering an altitude up to 10 km.

Indeed wind tunnel measurements may provide insights to degrading effects of the installation approach, but one has to be aware, that these tests are not replacing in-flight calibration of e.g. the wind measurements.

The sensor package is now discussed much more in detail (Subsection 2.3), esp. Figure 4a. Additionally, we refer to articles of ours where the same sensor type was installed and used (line 247).

For example, when the TAMDAR system was developed, questions regarding biases produced by instrument lag using capacitive humidity sensors flying through the atmosphere were sufficient to change the instrument design to use dual sensors whose reports were averaged.

While the sensor lag of capacitive sensors is not eliminated by deploying two identical sensors (to provide redundancy as used in TAMDAR, see e.g. Mulally 2009 [6]), we agree that the complementary use of different sensor principles is of high advantage. Sensor readings might have quite complex transfer functions between the atmospheric state and the observation quantity. On current research aircraft, the approach is to combine sensors of different measurement principles with different advantages, e.g. a stable, high accuracy sensor with a sensor of drifting signal, but high resolution response time, by complementary filtering (high pass for the fast sensor, low pass for the slow sensor with suitable cutoff frequency).

We decided to add details about how to correct for sensor lag (Subsection 2.5).

I also expected to see some discussion about how wind measurements were derived from the aircraft and how they were affected by turbulence, especially near the level of the jet stream. Unlike other proven aircraft observing systems which attain speeds much larger than the ambient wind in the upper troposphere, the drone discussed here is likely to be flying at speeds much less than the atmospheric flow. As such, much of the wind vector determination will be dominated by change in geographical location.

Based on your comment, we decided to include more information about the wind derivation method, which enables further discussion whether wind speed and turbulence impact the accuracy of the wind vector calculation. In general, low air speed (flight speed) increases the accuracy of the wind measurements, as the vector difference (between the air flow vector and the inertial motion vector) benefits from small vector length assuming nonzero angular errors (here, the vectors are assumed to

be defined by angles and lengths). The wind vector is not derived from changes in geographical locations, as it is the case for the classical radiosonde.

Based on this comment, we included information about the derivation of wind observations (Subsubsection 2.5.3).

Without this documentation [or clear knowledge of the system outputs (e.g., GPS locations, drone speed and direction, maximum flight duration, maximum distance from takeoff site to tropopause observation, residence time at tropopause elevation, etc.) that are used to determine the meteorological parameters], it is difficult to assess either the utility of the drone observations or the source of differences between the drone observations and other sources.

We now state more clearly in the manuscript:
"The drone measurements can take place from any location, up to the altitude of 10 km, if flight permission is obtained. It is completely independent of additional infrastructure, like an airport, or the availability of helium. The drone only requires a cylindrical air space with a radius of about 11 km for the whole mission. The drone ascends and descends at a vertical speed of 10 m/s and a horizontal speed component of 100 km/h. Therefore, the mission up to 10 km altitude and back to the landing site takes in total 33 min. The data is available by remote transfer with a temporal resolution of 1 Hz. The full data set with a resolution of up to 100 Hz can be downloaded after landing, and is processed automatically for upload in GTS." During the finalization of the revised manuscript, the text might change slightly.

The text above is now included in line 410ff.

I would also like to have seen evidence that any object that is 2 m in length and 5-6 kg in mass will not be 'tossed around' in the jet stream. This would have to have a substantial impact of some wind observations, which would than need to be eliminated using specialized quality control.

The drone was designed for an air speed of 100 km/h, and it was shown that it is capable of handling this wind speed. As mentioned above, there is no particular impact on wind observations.

As this comment is likely based on misinterpretation with respect to aircraft platforms, we did not implement any changes besides the calculus for wind measurements (Subsubsection 2.5.3).

Similarly, since the authors state that the temperature reading may be affected by heat generated within the drone itself, is there a method of determining which readings are likely to be contaminated as a function of whether the drone is flying into the wind (in which case the heat generated by the drone should be moved away from the rear of the aircraft and away from the nose mounted sensors) or with a tailwind (in which case the heat could remain in the proximity of the drone and possible affect the sensors).

An aircraft with tailwind is flying through the air at the same true airspeed as it is with headwind, so this is clearly not the case.

As this comment is likely based on misinterpretation with respect to aircraft platforms, we did not implement any changes.

Two other sections of the paper provide insufficient support. In the 20 lines of the System Design section, the authors use Figure 1 to provide typical ranges of temperature and wind conditions that were used in the drone design. Panel 1b clearly shows that more than 10% of wind observations at 9-10km exceed 40m/s, yet the instrument is designed to operate in wind speeds < 28m/s. That amounts to a difference in Kinetic Energy of 50%. Significant weather events (e.g., cyclogenesis) are very often associated with very strong jet streaks and substantial ageostrophic circulations that enhance the environment in which the storms form. Accurate knowledge of the sources of Kinetic Energy and its conversion into Available Potential Energy is essential to improve forecasts of these events. If the objective of launching drones is to fill in observations at times when are expected, it is unlikely that the existing design limits of this system will fill that need. This limitation of environments in which wind

speeds are < 28m/s makes the use of the phrase "up to the tropopause" inappropriate for inclusion in the title.

This was written misleadingly. The system is designed to climb up to the troposphere even during a constant wind field with FF=28m/s – BUT typically, the wind exceeds 28 m/s only in some altitudes, which enables the UAS to gain ground (fly into the direction where the wind comes from to generate some buffer) before entering the high wind speed.

The aim of developing the drone was to complement regular radiosondes. The focus is not on chasing storms or situations of particularly high wind speed. However, also radiosonde ascents are limited, and launching a radiosonde is only possible up to a surface wind speed of around 20 m/s.

Kinetic Energy is not a suitable concept for describing the developed drone, as only a small amount of Electrical Energy is needed to gain the Kinetic Energy – the mission is dominated by Potential Energy needed to climb up to 10 km, with increasing Frictional Energy (aircraft drag) at high wind speed, when the aircraft is forced to climb at a higher horizontal air speed than 28 m/s.

We included the potential availability using the "integral wind speed" in 176ff.

Also, the authors seem focused on moisture in the middle-upper troposphere, saying that it is particularly important. I'll grant that moisture can have a number of influences at these levels, but the amount of moisture, its vertical structure and horizontal structure are much more important for forecasting weather events from heavy precipitation and flooding to severe storms.

We agree that the amount of moisture, vertical and horizontal structure is of high importance for weather. However, in our study we focus on the potential benefit of additional radiosonde-like data up to the tropopause, whereas prior studies seem to focus more on the boundary layer. Therefore, we analyze in the study at which altitudes there are typically changes at which time scales. In contrast to the upper troposphere, which changes more slowly, and the atmospheric boundary layer with a pronounced diurnal cycle, there is the region in the middle-upper troposphere with irregular, but frequent changes of the meteorological parameters, in particular moisture, which could benefit from additional data.

- no changes within the manuscript based on this comment -

The section entitled "Potential for covering data gaps with UAS based on atmospheric dynamics" seems mistitled and contains substantial conjecture. Figure 6 provides climatological, not dynamical, information about the rate of change of atmospheric parameters, not the dynamical causes for these changes. It is not new and not particularly relevant to the results presented here, which should be the emphasis of this paper. For example, if the drone system is limited to elevations below ~10km, why to these plots extend up to 35km and how is the reader expected to extract information from them? In addition, phrase like "As one would expect" and "likely results in" are unsubstantiated by the results presented in the paper and detract as conjecture.

As we present small UAS as a potential brick to be part of the Global Observing System, the intention is to show the variability of the atmosphere covering altitude up to the average ceiling of radiosonde measurements. The low variability above the Troposphere/UTLS is also represented by decreased accuracy and spatial resolution breakthrough requirements provided by WMO OSCAR [7].

We moved the section into "Discussion and Conclusion" and substituted "dynamics" by "variability". Soft expressions such as "likely results" were removed, and further information how to interpret the Figure is added.

We moved the text into Subsection 2.7 and referred to within the discussion (516ff)

My biggest issues with the paper, however, relate to the lack of any quantitative assessment of the drone observations or their possible impact on operational weather forecasting.

The focus of the study is to present the new drone system with the capability of reaching an altitude of 10 km. Data gathered by LUCA will be implemented in numerical weather forecast in future studies.

Nevertheless, we decided to analyze the measurements gathered with the simplistic sensor setup as a case study with methods similar to the methods used in Wagner and Peterson 2021 [1].

Data is now assessed in a quantitative way within Subsection 3.3 (esp. Figure 8 and Table 2) revealing the potential impact on operational NWP.

First, neither Figs 4 nor 5 indicate the units of the wind observations. Are these m/s or knots? Because some LUCA reports in figure 5 show a filled barb on the wind flag, I can only assume that they are knots.

We now added the unit of wind speed in the figure captions.

Units added in Figure 6 and 7

Figure 5 provides anecdotal information from 2 radiosonde matchups that can only be used subjectively.

The figure shows that radiosondes provide similar data. Of course a direct intercomparison is difficult due to the different methods, and in particular the increasing distance between the two systems. We removed the Panel 5b) and added more information about the comparison within the case study "Assessment of data quality using a simplistic sensor setup on the platform LUCA"

Using a method to find collocated measurements (radiosonde and drone) similar to [1], we were able to assess data quality quantitatively (Method in Subsection 2.6, Results in Subsection 3.3)

Figure 4, however, indicated that colocation information was available from 5 launches, yet no attempt was made to quantify the differences. In several instances, the authors attribute differences between the two observing systems to time mismatches between the sonde and the drone.

The colocation was redetermined according to the technique used in [1] to intercompare data only within a spatial and temporal limit of 50 km and 30 min. The flights were analyzed in the new subsection "Assessment of data quality using a simplistic sensor setup on the platform LUCA".

Now a (successful) attempt to quantify the differences is made (Table 2).

Articles assessing the accuracy of AMDAR observations against a longer series of special radiosonde launches prior to and after multiple AMDAR aircraft ascents/descents showed that the time difference effects were substantially less than other factors.

We agree that this is an important point; however, it is beyond the scope of this article.

- no changes within the manuscript based on this comment -

When enlarged enough to view the individual wind barbs in Panel 5b, wind speed differences of up to 15 knots are noted at multiple levels around 6km, with erratic directional behavior in the drone observations near 2km and between 4 and 6 km. There is also little evidence in Fig 4 to justify the 2-3 degrees warmer observations near the top of the drone profile in Panel 5b. These differences are crying for detailed quantitative analysis and explanation, but none is available.

We agree that the measurements require a detailed quantitative analysis and explanation in the future. Panel 5b) was removed, as no simultaneously launched radiosonde is available. A quantitative analysis of the data was included as a subsection in "Results".

However, the main aim of this article is to demonstrate the success of developing a drone capable of flying up to the tropopause. The sensor package that was installed was mainly for demonstration purposes. With more time, and with the experience gained with the drone, a more sophisticated

sensor package will be installed. The focus of the study is the carrier system, the drone, and with the preliminary measurement package, the link towards the use in operational meteorology is indicated.

Based on this comment, the measurements were corrected for an assumed heat transfer within the fuselage (2.5.1, line 314ff).

The paper is also devoid of any cost/benefit analysis for the use of a system like this in daily operations. After all, the best system in the world will be of little use if weather services can't afford to use it. What is the cost of producing one set of ascent/descent reports? How does this compare to existing radiosonde system costs? (FYI, the NWS in the US has done a detailed analysis of this and find the total cost of radiosonde launches to be in the range or $200 to $250 per launch, personnel, instrument, balloon, and gas.)

An attempt to quantify the cost and compare to conventional radiosonde is done in Bärfuss et al., 2022 [2]. We included some sentences about cost in the conclusion/outlook.
As a research article to describe the feasibility of such measurements, cost/benefit analysis have been left out, and commercial vendors would adjust the system according to the market. Taking into account the negative environmental impact of radiosondes (which might be considered substantially different), a cost/benefit analysis relies rather on politics than on training/operating/system cost.

As the focus of the article is on the technical feasibility and data quality of drone measurements, only a brief note about cost was included in line 604ff. We additionally refer to the technical article [2], where some cost estimation is shown.

How timely is observation availability, especially if they are to be used on the mesoscale?

In the article we describe the procedure of using the drone. It can be launched any time. Quicklook data are available during the flight. The full data is available after landing, which is 35 min after takeoff. Then the data is then quality checked, transcoded and transferred to the GTS within less than 3 hours after takeoff. Targeted timeliness is 30 min.

We added information about timeliness in line 220/221 and 410ff.

Are the observations 'all weather', e.g., can the system fly through condition of aircraft icing? If not, the system will become less attractive.

In the article we mention the ability of flying in icing condition and the concept briefly. For some more information, please refer to the technical article on the system LUCA [2].

We now refer to the all-weather strategy of the drone LUCA described briefly in [2].

How quickly can the drone be recharged and sent to another mission?

The turn-around time between landing and the next launch is around 20 min using multiple battery packs and replace them during the ground cycle. This is now stated in the manuscript.

See lines 415/416

What a training is needed both 1) to launch/retrieve the system and process the data

As the system performs the whole mission automatically, the operator only has to be familiar with the launching and landing mechanism, and can control the performance of the system at the operator station. If problems are detected, e.g. lower climb rate due to icing, the mission can be aborted. Data processing is highly automated. However, this is still a system under development, and not yet a product available on the market.

No changes within the manuscript based on this comment, except a brief note in 604ff.

and 2) maintain the system?

As the maintenance requirements depend on the authorities and the granted permission, we now describe the expected maintenance routines needed in the conclusion/outlook.

No changes within the manuscript based on this comment, except a brief note in 604ff.

How many systems will be needed to fill the needs of the EU?

As the radiosonde net is quite dense in Europe, the system might be more beneficial for other sites. In particular flight permissions may be difficult to obtain in highly populated areas than in other places, e.g. the Antarctic. The system was designed to be operated in the Antarctic environments, as there are less frequent radiosonde launches, and the environmental aspect is of high importance there.

- no changes within the manuscript based on this comment -

As a result of inadequacies like these throughout the text, it read like a preliminary progress report instead of a quantitatively verified scientific contribution to the literature. Although I support concept presented here, the paper is not ready for publication at this time.

Thank you for your valuable comments and for sharing your perspective. We see that the paper is lacking a data analysis of the measurements (despite the simple sensor setup) as well as strong statements about the focus of the study, which might have distracted you from the main utterance - the feasibility of using small drones up to the troposphere as carrier systems for atmospheric observations.

[1] Wagner, T. J., & Petersen, R. A. (2021). On the Performance of Airborne Meteorological Observations against Other In Situ Measurements, Journal of Atmospheric and Oceanic Technology, 38(6), 1217-1230. Retrieved Jan 10, 2023, from
https://journals.ametsoc.org/view/journals/atot/38/6/JTECH-D-20-0182.1.xml

[2] Bärfuss K, Dirksen R, Schmithüsen H, Bretschneider L, Pätzold F, Bollmann S, Panten P, Rausch T, Lampert A. Drone-Based Atmospheric Soundings Up to an Altitude of 10 km-Technical Approach towards Operations. Drones. 2022; 6(12):404.
https://doi.org/10.3390/drones6120404

[3] Bärfuss K, Pätzold F, Altstädter B, Kathe E, Nowak S, Bretschneider L, Bestmann U, Lampert A. New Setup of the UAS ALADINA for Measuring Boundary Layer Properties, Atmospheric Particles and Solar Radiation. Atmosphere. 2018; 9(1):28.
https://doi.org/10.3390/atmos9010028

[4] Moninger, W. R., Benjamin, S. G., Jamison, B. D., Schlatter, T. W., Smith, T. L., & Szoke, E. J. (2010). Evaluation of Regional Aircraft Observations Using TAMDAR, Weather and Forecasting, 25(2), 627-645. Retrieved Jan 10, 2023, from
https://journals.ametsoc.org/view/journals/wefo/25/2/2009waf2222321_1.xml

[5] Amdar reference manual--aircraft meteorological data relay. (2003). World Meteorological Organization. Retrieved Jan 10, 2023, from
https://library.wmo.int/doc_num.php?explnum_id=9026

[6] Mulally, Daniel. (2009). The TAMDAR Sensor's Relative Humidity Performance on ERJ-145 Commercial Aircraft. Retrieved Jan 10, 2023, from
https://ams.confex.com/ams/pdfpapers/149882.pdf

[7] WMO OSCAR
https://space.oscar.wmo.int/requirements

---

## Referee Report (RR1)

As I said in my previous review, I am a strong supporter of the utility and benefit of automated aircraft reports to fill the time and space gaps left between other in situ observing systems. Upon rereading the article "Drone-based meteorological observations up to the tropopause a concept study" by Barfüss, Schmithüsen and Lampert, I feel that it is now ready for publication after a few minor changes. Some of my reasoning for this decision for the suggested changes is summarized below. I apologize for some possible disjointedness in the review, but that may in places reflect the need for more organization in the paper's organization.

To start, I thank the authors for their continued efforts to improve the paper. The current version is much more informative and leave many fewer points unresolved.

The introduction remains essentially unchanged and includes very little discussion of problems that other authors have documented with automated reports from commercial aircraft that should provide a far more stable platform for collecting data and providing representative measurements or the atmosphere without being affected by artifacts related to aircraft stability. For example, AMDAR wind observations from longer-range aircraft are much more accurate (by a factor of 2) than TAMDAR reports obtained from smaller/lighter regional jets. Possibly more distracting to is the statement in the first sentence of the Abstract reads "with large data gaps in the atmospheric boundary layer, above the oceans and in polar regions", with only a brief mention of that "the feasibility of reaching an altitude of 10 km with a small meteorologically equipped drone is shown." Through the remainder of the text, the utility of the drone observations, however, is judged by the ability to reach 10km in a polar environment. The Abstract should be revised accordingly.

Grammar and word choice continue to present problems throughout the paper. For example. In line 77, the term "Breakthrough requirements" is not defined. Are these "new' requirements or mesoscale requirements that were never intended to be fully met by GOS. Although lines 77-83 attempt to clarify this issue, inferring that 100 km horizontal and 1 km vertical resolution will not be sufficient to meet also local and regional needs, especially without a clear statement of temporal frequency.

In line 89, the term "back in the 60ies" should be "60s", or better should be simplified to "at that time". Also, in line 89, replacing "using binoculars" with "using theodolites" would suggest that the observations could be quantitative, not just qualitative.

Lines 98-105 – The idea of using this application in the future is intriguing, but it must be noted that it will only have benefit if the full-resolution data can be downlinked and distributed in real time, which can't be done with the current system.

Line 122 – Add "extremely low specific humidities" to the list of challenges after "temperatures"

Line 131 – Replace "to play . . . and" with "as a future alternative to"

Line 156 – Replace "for a time period of" with "episodically over"

Line 167 – Be more direct at beginning of sentence by eliminating "trade-off" notion with "The LUCA system was designed . . ."

Line 171 – Replace 'Applying wind speed condition" with "Restricting wind speed conditions"

Line 178 – Are you saying that LUCA will operate at times when radiosondes will fail, e.g., in conditions of "rainfall, snow, heavy turbulence and within clouds". No proof for that statement is shown, unless drones were launched alongside every radiosonde launch. This needs to be clarified.

Line 190 – The fact that 45% could severely reduce the number of drone profiles that are available, making the system appear to not be "all weather". Can you estimate how many LUCA flights would be missed due to the potential for icing?

Line 198 – Add "shown here" after "flights" and be clear whether an icing sensor was in fact installed, not just "prepared to be installed."

Line 220 – It would be good to include the typical ascent rate values along with the other performance specifications here so the reader doesn't have to hunt through the paper to find them

Line 236 – What is the meaning of the first sentence? It could be read to imply that multiple sensors were used for each atmospheric parameter being measured. Eliminating it removes nothing of value from the text.

Line 241 – Mention should be made both that the radiosonde sensors were designed to have a low ventilation rate as the balloon drifts with the wind and that a drone travelling at constant air speed of 28 m/s will have much greater ventilation rate. How is these accounted for? If this is discussed later, say so.

I will stop with individual comments here but suggest that the authors review the text thoroughly for other wording errors or inconsistency, if any. Overall, the details of the presentation have improved significantly since the first version.

Lines 290-564 – The authors have provided a very good description of the data processing and error estimation procedures. While the information is extremely helpful, I found that the amount of detail presented distracted the reader from the flow of the paper's primary messages. As such, I highly recommend that this portion of the paper be moved to an appendix dedicated to the subject.

Line 334 – Are the coefficients the same during ascent and descent? Are these results consistent with documented hysteresis issues using radiosonde-like sensors for TAMDAR aircraft? Table 2 seems to imply that there is no hysteresis during ascent with a warm bias during descent, which is opposite of the TAMDAR results. How much is this a function of the choice of $M \sim 0.07$ in Equation 5?

Line 451 – If the time constant for temperature reports is 21 seconds, please show how that translate into spatial and vertical averaging distances.  E.g., does 21 seconds for a drone travelling at 28 m/s (and 10 m/s vertically) in still winds conditions equate to 0.588 km horizontal averaging distance and a 210 m vertical average (~21 hPa near the surface).  How would this change with tailwinds and headwinds?

Line 460 – For RH, the question becomes more complicated.  For the same still conditions, do3s this mean that the spatial and temporal smoothing at the surface are 0.420 km and 150 m (~15 hPa) respectively, which at 3 km elevation become1.68 km and 600 m (~40 hPa) and at 10 km are longer than the entire flight length (2000 s = 33.33 minutes), therefore making the RH values at high levels reflections of the average RH throughout the entire flight.

---Please include vertical profiles of the horizontal and vertical averaging distances that have been applied to each type of observation somewhere in the text.  Figure 11 might be a candidate location.

Figure 5 and text that goes with it.  – I like this example.  Thanks for creating it.

Line 664 – Clearly state the data set details.  As it stands, I don't know whether the statistics were obtained from the profiles in Fig. 9 or from all profiles in Fig. 8 until the second paragraph into the section.

Line 666 – The word "treats" is clearly not correct.  Should this have been "threats" or better yet, "factors"?

Figure 10, center top panel (RH) – A shift of the apex of a probability density plot away from zero (in this case toward -2.5%) usually reflects a bias in an observing system.  Please include a statement of how the variable time averaging that was applied in line 460 affected this.

Table 2 – Please ad the number "6" before "ascent" and "descent" in the last sentence of the caption.  It makes it easier for the reader to understand the sample size.

End of paper – I think that it is still important not to ignore the issue of system and operations costs.  That could be incorporated in the future, more extensive testing that was suggested.  Because much of the thrust of the paper was addressed at substituting for radiosondes, lack of mention of the question weakens what has become a very good paper.

As I said in earlier reviews, I support concept presented here.  With the enhancements that the authors have provided, the paper is now essentially ready for publication, after the small number comments are addressed.  Again, my intent here has been to provide constructive suggestions for improving the paper.  I do not need to see the final revised version.

I look forward to seeing the article in print – Ralph Petersen

---

## Author Response (AR2)

**Author Response Revision2 - amt-2022-236**
Drone-based meteorological observations up to the tropopause
Konrad Benedikt Bärfuss et al., Atmos. Meas. Tech. Discuss.
https://doi.org/10.5194/amt-2022-236, 2022

Dear Editor,

In the following, you will find a list of the referee comments and all associated changes within the manuscript in addition to the diff-file. Please note that line numbers of changes based on reviewer comments are now associated to the plain revised document (not the diff-file).

As a major change, you will now find a well-organized uncertainty discussion starting with details on the correction algorithms. As these techniques are applicable to other drone-based meteorological observations, we think that the value of the additional sections more than compensates the increased length of the article.

We hope that the quite brief notes on what changes were initiated by which of the reviewer's comments are in your interest.

Now let's start:

**Referee Report #1**

The Authors would like to thank the referee for taking the time to again review our work.

The main criticism of my first review was the lack of a proper sensor description and validation. This includes an error estimation, as the sensors have to give reliable readings under extreme conditions during the flight, including a wide range of temperature, humidity and wind speeds, which is usually not the case in the atmospheric boundary layer. Unfortunately, from my point of view, this is still not adequately and sufficiently addressed in the revised version. However, I expect this for publication in a journal for atmospheric measurement techniques.

We now added a detailed description of postprocessing methods in Sect. 2.5, an explanation to the calibration techniques in Sect. 2.6, and a dedicated error discussion in Sect. 2.7.1.

Furthermore, the new sections implemented to compensate for measurement errors contain equations that are inadequately presented. Moreover, these corrections are not adequately validated.

We intensely overworked and completed the presentation of the measurement error compensation and other post processing techniques which now is a substantial part of the article (see above).

E.g.
• Sect. 2.3 makes assumptions about the inaccuracy of the wind estimation, but there is no systematic analysis or error estimate. Further, in Sect. 2.5.3 a transformation of the aerodynamic coordinate system to the geodetic coordinate system is not appropriately demonstrated, and simplifications and speculations are made that are not comprehensible to me as a reader.

A systematic analysis and a complete description of the wind estimation algorithms including simplifications (also those applied for AMDAR) and their impact on the results are now presented in Sect. 2.5.3 and applied on the drone measurements in Sect. 2.7.1.

• In Sect. 2.5.1. a temperature correction is carried out. This should also contain the dynamic pressure (air speed).

We now recap the standard approach for the correction of the temperature rise caused by compressible effects and discuss the error when incompressibility is assumed (as it is done in this article) in Section 2.5.1.

• Section 2.5.2 introduces an approximation using a power law and specific estimates however, a comprehensible determination of the parameters is also missing here.

The additional Section 2.6 now introduces the calibration/validation method used to estimate the parameters for power law representing the time constant for humidity readings.

Minor remark: In a scientific publication, the official term uncrewed aircraft system (UAS), which legislators and the ICAO use, should be used rather than the still colloquial and military term 'drone'. Using 'uncrewed' also enables the term to be gendered.

Unfortunately, there is no common agreement on the terms to be used for unmanned aerial systems. EASA, ICAO and WMO use the term UAS or even wxUAS for weather sensing UAS (Pinto et. al. 2021). In order to make the article findable by a community searching for other terms, we now at least introduce the term UAS: "Drones (also called unmanned or uncrewed aircraft systems, UAS, or remotely piloted aircraft systems, RPAS)"

**Referee Report #2**

The authors would like to thank the reviewer for the detailed comments and the interesting discussion. In the following the raised issues are answered point by point. In case line numbers are mentioned by the authors, they refer to the revised version of the manuscript (without track changes).

As I said in my previous review, I am a strong supporter of the utility and benefit of automated aircraft reports to fill the time and space gaps left between other in situ observing systems. Upon rereading the article "Drone-based meteorological observations up to the tropopause" by Barfüss, Schmithüsen and Lampert, I continue feel that it is not yet ready for publication. Although I appreciate that the authors have attempted to address some of my concerns, the text continues to read much more like a "Concept Study" or perhaps "A Report on Preliminary Project Activities" than providing significant scientific study that would support publication on scientific merit as it stands. As such, it needs to be revised and improved before it can be accepted for publication. Some of my reasoning for this decision is summarized below. I apologize for some possible disjointedness in the review, but that may in places reflect the need for more organization in the paper's organization.

We consider the fact that we reached the altitude of 10 km with such a small system and own propulsion a major step towards expanding future radio sounding capabilities based on drones. This provides new opportunities for the branch of meteorological measurements, and therefore we would like to present this new tool to the scientific community. As far as we know, this constitutes even a world record. In addition to this mainly technical achievement, we even included first simple meteorological instrumentation to demonstrate the scientific benefit of this kind of data.

To start, I suggest that the authors retitle the work to include the words "Concept Study" in the title. That will make it absolutely clear to the reader that they should be expecting to see preliminary results, not extensive validation.

As suggested, we changed the title to "Drone-based meteorological observations up to the tropopause - a concept study".

The introduction remains essentially unchanged and includes very little discussion of problems that other authors have documented with automated reports from commercial aircraft that should provide a far more stable platform for collecting data and providing representative measurements or the atmosphere without being affected by artifacts related to aircraft stability. For example, AMDAR wind

observations from longer-range aircraft are much more accurate (by a factor of 2) than TAMDAR reports obtained from smaller/lighter regional jets.

We consider that the AMDAR/TAMDAR measurements are not the object of our investigations, and therefore would prefer not to mention too many details in the introduction. The link to aircraft stability is misleading, as artefacts are seen to be caused by the inertial navigation system in literature, see Moninger et al. 2010, which states that "The lower quality of the wind data from TAMDAR is likely due to the less accurate heading information provided to TAMDAR by the Saab-340b avionics system. Accurate heading information is required for the wind calculation, and the Saab heading sensor is magnetic and known to be less accurate than the heading sensors commonly used on large jets."

Possibly more distracting to is the statement in the first sentence of the Abstract reads "with large data gaps in the atmospheric boundary layer, above the oceans and in polar regions", with only a brief mention of that "the feasibility of reaching an altitude of 10 km with a small meteorologically equipped drone is shown." Through the remainder of the text, the utility of the drone observations, however, is judged by the ability to reach 10km in a polar environment. The Abstract should be revised accordingly.

We agree that this is misleading, and revised the first sentences of the abstract. The text in the abstract has been changed to:

"The main in-situ data base for numerical weather prediction currently relies on radiosonde and airliner observations, with large systematic data gaps, horizontally in certain countries, above the oceans and in polar regions, and vertically in the rapidly changing atmospheric boundary layer, but also up to the tropopause in areas with low air traffic."

Grammar and word choice continue to present problems throughout the paper. For example, the end of the sentence that reads "whereas data from level flight is . . ." in line 49 of the manuscript is confusing to read. Using "with additional observations provided at flight level" would have read more clearly and been more concise.

We changed the text in l. 50 to "with additional observations at flight level provided during cruise".

This is one of many places where the text should be reviewed grammatically using a tool like the "Editor" included in Microsoft Word.

We would like to thank the reviewer for suggestions for improvement of language, and consequently checked the text using language tools for both spelling and grammar.

This should help improve the conciseness of the text and suggest more appropriate word choices in places. The authors also often incorrectly use the noun "data" as being singular (followed by 'is' instead of 'are").

We now use data as a plural noun throughout the text as preferred in general scientific writing.

As further point of confusion for me in reading the author's response to the first review are the errors line numbers noted by the authors. For example, a refence is made early in the response to my review to section line 232 in Section 2.5.3, but that section only extends from lines 314 to 335. The reference about AMDAR/TAMDAR in line 344 can't be found (except possibly much later in line 446). This line numbering inconsistency is present at other points in the paper as well, making it harder to review.

The line numbers change with the revision of the text, of course. Our line numbering referred to the document with the tracked changes. In the current review we refer to the plain revised manuscript, not the track change document.

I thank the authors for focusing on using "drone" and "LUCA".

Unfortunately, there is no common agreement on the terms to be used for unmanned aerial systems. EASA, ICAO and WMO use the term UAS or even wxUAS for weather sensing UAS (Pinto et. al. 2021). In order to make the article findable by a community searching for other terms, we now at least introduce the term UAS in l. 86: "Drones (also called unmanned or uncrewed aircraft systems, UAS, or remotely piloted aircraft systems, RPAS)"

Please use metric system measurements throughout the paper, removing conversions to other units (such as kn).

Thank you for insisting on this – we were not aware of the fact that wind barbs are actually also defined in m/s by the WMO (Manual on Codes) and removed the unit "kn" throughout the manuscript.

The term "integral mean wind speed" in line 167 needs to be defined. Also, if the purpose of this restriction is so that "the drone does not have to stay above the launch point, but must be able to return to the base" as stated in line 170, isn't it more important that the total vector displacement of the drone from its launch site be limited, not just the mean wind speed? Please explain further and restate what this means about limits on wind speeds that will not be observed. This needs to be clearly referenced again at the end of the paper under limitations.

We changed the text in l. 167 to: "As a trade-off for LUCA, the system was designed for operation in the temperature range between -75°C and +30°C, and for a wind speed of less than 28 m/s over the whole vertical profile to limit the maximum vector displacement of the drone from its launch site."

Line 171 is changed to "Applying wind speed conditions such that the drone does not have to stay above the launch point, but must be able to return to the base, the mean wind speed over the atmospheric profile to be observed by the drone should not exceed the nominal horizontal airspeed component of 28 m/s."

In the conclusion section, l. 743, we added: "The limitations concerning wind speed are related to the current maximum airspeed of 50 m/s."

Lines 154-159. Figure 1 should be discussed after the integral speed limit is defined and explained, not before.
Although the minimum takeoff speed and ascent rate of the drone are mentioned, but it is not easy to find the average airspeed of the drone. Please provide this, along with how greatly and rapidly it varies during flight.

Figure 1 was our starting point for designing the aircraft. We used the statistical analyses of environmental conditions for the two sites Neumayer and Lindenberg in order to see what temperature and wind speed range the system has to endure to be competitive with radiosonde launches. We now state at the beginning of the Section 2 (l. 148):

"In the following, the process towards the design of the drone of type LUCA is presented briefly. Requirements for the system are derived from the environmental conditions to be expected, to obtain a high availability of measurements. Based on the environmental conditions for two sites, the design of the mission and of the drone is introduced. The simplistic sensor package that was used for the demonstration flights is described, including uncertainty aspects. The process of obtaining flight permissions for such altitudes is presented. The methods for data post processing are presented, and the obtained data quality is assessed. The section closes with a note on the variability of the atmosphere."

Concerning airspeed, we included in the text (l. 218): "As the result of a multi-variant optimization for profiling the atmosphere vertically up to 10 km, the design weighs 5-6 kg, depending on the deployed sensor package, has a wingspan less than 2 m, and operates at a constant airspeed of 28 m/s controlled by the autopilot system."

Table 1 is still not referenced in the text. It should be and it also be supplemented to include information about reporting frequency and representativeness and drone flight speeds etc.

Thank you for mentioning the missing reference. We now include in the text (l. 236): "For the measurements, different sensors were applied. An overview of the sensors and their accuracy according to data sheet is provided in Table 1."

We state in Sect. 3.1 l. 625 the measurement frequency: "The data are available by remote transfer with a temporal resolution of 1 Hz. The full data set with a resolution of up to 25 Hz can be downloaded after landing and is pre-processed automatically for upload into the GTS."

As this is independent of airspeed, we do not think that airspeed fits in the table about sensor characteristics.

The text (and Table) should also be clearly stated that wind tunnel tests have not been conducted to assess and document whether the installation approaches would degrade the observations further. The reader can only assume that no such tests were done of the LUCA installation approach.

We changed the text in l. 236 to:

"For the measurements, different sensors were applied. An overview of the sensors and their accuracy according to data sheet is provided in Tab. 1. The placement of the sensors is based on the experience with similar drone-based systems (e.g. Bärfuss et al., 2018, Lampert et al., 2020, Bärfuss et al., 2021), and the sensor behaviour and possibilities of correcting e.g. sensor response time, are well known. The performance of sensors and the data quality is assessed directly from the atmospheric measurements, without conducting simulations or wind tunnel tests for the overall setup."

Additionally, we clarified the importance of inflight calibration as the most important part within the Simulation – Wind Tunnel – In-flight ecosystem in aerospace science in Sect 2.6.

Lines 201-212. Please define data what is precisely is meant by 'low frequency' and how this differs from the practical limits on reporting frequency of independent drone observations in the final meteorological products.

We changed the text (l. 208) to: "During the flight, 1 Hz real time data are available."

Later in the revised text, it is noted that the time constant for response time of the humidity sensors are "15 s at 0 km, 60 s at 3 km and 2000 s at 10 km". Because 2000 seconds (33 minutes) is longer than the entire flight, does this mean that the moisture data in the cold half of the sounding may in fact represent only 1 deep layer of the atmosphere layer, even though the profiles shown in Figure 7 show more detail than this? Please explain how the detail can be accurate if the response times are so long. This could explain the differences noted near 550 hPa and above in both profiles in Fig7.

In fact, an immediate response will be measured even with such large time constants, which follows a fundamental law in sensing (and general exchange processes), but the value of the measurement is affected by the time constant. We now recap the control theory basics in Sect. 2.5 and describe the time lag correction of humidity in detail in Sect. 2.2.

The authors then downplay the importance of moisture observations in NWP improvement, sighting results from a global model (and possibly for longer forecast ranges).

We are not sure which section or sentence the reviewer is referring to. Maybe the following part of the manuscript (l. 778): "The quality of drone observations for sampling the complete troposphere is of superior importance, and could possibly contribute to climate applications. Regarding the involved variables, one finds that climate users tend to focus on temperature and humidity data (Ingleby et al.,

2022). For NWP applications, wind has arguably more than twice the impact on the quality of short range forecast (Ingleby et al., 2021)."

It is not our intention to downplay the importance of moisture observations. The NWP applications mentioned in Ingleby et al. (2021) refer to data denial tests in the ECMWF data assimilation system, and we changed the phrase in l. 782 to "For NWP applications, wind observations have arguably more than twice the impact on the quality of short range forecast compared to temperature observations (Ingleby et al., 2021)."

Similar data denial tests using other models (global and regional) show notable improvement is the shortest forecast times. That should not be ignored.

It would be great to get the references that are mentioned here, and then we can take them into account in the manuscript.

Section 2.5.3 needs to include much more information about the quality of the measurements going into the wind calculation, such as the frequency/precision of the drone location information (and derived drove velocity vector) along its flight path, the frequency and precision of the true air speed vector relative to the drone, along with a sample how these factors can affect the wind reports.

We now dedicate new sections to the calculation of the wind (Sect. 2.5.3) and error estimation (Sect. 2.7.1).

I was intrigued by the opportunity presented by the flight paths shown in Figure 4. It would be very instructive to study the variability of the meteorological observations during the periods of circling flight. This should be easy to do. The variability of the derived meteorological products during these loops should be quite consistent and seeing the error structures within the sets of drone reports could provide useful insight into the expected data quality.

We totally agree on the idea that measurements shall not correlate with the flight direction during e.g. a circle in constant altitude (neglecting turbulence). This technique is often used for in-flight calibration of sensors (even in geophysics). Upon your comment, we decided to mention/recap inflight calibration/validation techniques in the new Section 2.6.

For the drone measurements presented in the manuscript, comparing ascents with descents is the preferred method for the calibration/validation of temperature and humidity. Wind speed unfortunately is corrupt during flight phases where the engine was used (magnetometer data are distorted by the electrical current), which was also the case during the circles you mention. Therefore, ascent wind data (generally data under "power") are regarded as corrupt and not discussed further, except the illustrative statistics in Table 2.

Errors from factors discussed in lines 244-250 should also be noted in discussion about Table 1.

We now elaborate theoretical errors in Sect. 2.7.1 in detail and decided to keep Table 1 as simple as possible.

The authors now reference work by VAISALA (2021) regarding ventilation effects and instrument response lags as a basis from their temperature and humidity corrections. At no point, however, is the validity of the Vaisala method us affected by the increased ventilation that could be present in a drone moving both vertically and horizontally through the atmosphere at higher speeds than a drifting balloon. This shouldn't be ignored.

We take this comment as indication that we did not properly introduce the control theory essentials for this correction, and dedicate the first part of Sect. 2.5 to the underlying fundamentals of sensing and time response.

I also expected to see a more quantitative discussion of how the wind measurements were derived from the aircraft and how they were affected by turbulence, especially below and near the level of the jet stream, along with a more detailed error analysis.

We now dedicate a whole subsection (Sect. 2.5.3 to the wind calculation, which is based on the vector difference between the airspeed and the ground speed. The wind measurements are not affected by turbulence, instead, turbulence parameters can be calculated from the data.

I also still don't see any evidence that any object that is 2 m in length and 5-6 kg in mass will not be 'tossed around' in areas of large shear and turbulence near the jet stream and in the boundary layer.

"Tossing around" is a quite illustrative term, so let us argue less scientifically: We don't see any evidence why the aircraft should be tossed around. Regarding dynamic soaring with unmanned aerial vehicles (3 m length at 9 kg mass, similar wing loading) in extreme wind shear and heavy turbulence (reaching an airspeed of the UAS over 500 mph at over 50 mph wind speed, which albeit is a significantly lower wind speed than to be expected within a jet stream core), you might be convinced by this video that drones are not "tossed around": https://www.youtube.com/watch?v=GCVK3w5DHbk

Similarly, since the authors state that the temperature reading may be affected by heat generated within the drone itself, I'm still not convinced by the current text that the readings are likely to be contaminated as a function of whether the drone is flying into the wind (in which case the heat generated by the drone should be moved away from the rear of the aircraft and away from the nose mounted sensors) or with a weak tailwind (in which case the heat could remain in the proximity of the drone and possible affect the sensors).

We disagree on this as an aircraft is flying with a certain speed with respect to the air (completely decoupled from the wind which only affects the heading of the aircraft to reach a certain earth-fixed point) and do not know how to resolve the misconception here.

As noted in the previous review, Section 2.7 and Figure 5, though of some interest, do not add to the substance of this paper and should be removed, especially because the data used are at time and space scaled substantially later than those of individual drone flights.

We do not agree on this point, as the figure is not intended to show variability during one flight, but variability within a day – enabling a discussion of the benefit of multiple launches per day. It might be obvious for scientists working on e.g. atmospheric modelling and data assimilation that the variability of the atmosphere decreases above the tropopause (except the diurnal cycle for e.g. temperature is usually well described by models), but not necessarily for others working e.g. in the field of drone- and sensor design.

Also, please remove the ERA5 curves from Fig 7, since they do not add value to the discussion.

The idea of our measurement system is to provide data for the modelling community. As ERA5 has the capability of highly flexible data assimilation, the drone data could be easily integrated here. Therefore we would like to show the comparison. We do consider it interesting that the T profiles agree rather well with radiosonde and drone data, but the altitude and strength of temperature inversions within the boundary layer is not captured properly. Differences in the mixing ratio are more pronounced.

It would be more valuable to compare the differences/similarities between the LUCA profiles obtained during ascent with those obtained during descent.

We agree regarding the parameters T and RH. For wind, the ascent is not regarded as an observation due to (expectable) magnetic interference. We clarify this now in Table 2 and in l. 679 ff.

Finally, summary statistics on the differences between the LUCA profiles and each radiosonde data set should be included on each panel.

We now add the error statistics in the panels of Figure 11 along with the estimated uncertainties.

Also, in Figure 7, winds are plotted about every 300 m in height. Is this the how frequently they are available? If there were more drone wind report available, what was the variability of the drone winds between the reported levels?

We now mention the wind observation frequency (25 Hz) in l. 625. The variability of the wind measurements between the reported levels (1 m resolution) is graphically shown in Figure 11.

Regarding Figure 8 and supporting text, please explain what is meant by "virtual air parcel position was determined by virtually back-shifting the probed air parcel" in the figure caption, especially since the term 'back-shifting' if not used in Section 2.6 as suggested in the text.

We changed the text in line 570ff that this is now accessible.

I commend the authors for adding and a discussing Table 2. However, it is not clear stated that the histograms used all drone reports or only those shown in Figure 7, nor does it explain why there are twice as many data matchup for descents as for ascents, especially since the ascent and descent rates were said to be similar and since they suggested earlier that quality control limits were not imposed on the drone data sets. Also, it would be good to include gross statistics including both ascent and descent matchups. Finally, the more negative (less positive) biases shown during ascent are opposite from what would be expected due to hysteresis effects.

The hysteresis is completely cancelled out by the time lag correction, which is now discussed in detail in Section 2.5. As the matchups follow the launch times of the radiosondes in respect to the drone trajectory and the wind, we omit the explanation of the data matchup numbers and take it as descriptive values. An elaboration of these matchup numbers would be out of the scope of our manuscript.

Finally, the captions should describe the structure of the figures, not an analysis of their content. That should be in the text.

We moved the analysis part of Fig. 7 (previously Fig. 6) into the text section (line 593ff).

In addition, without a substantial greater number of further experiments and tests, the last sentence of the caption for Table 2 stating that "This indicates a possibly higher accuracy of the drone observations than observations within the AMDAR/- TAMDAR programme" is conjecture and must be removed.

We removed the sentence from the caption of Table 2 as suggested.

Though I'd expected to see mention of ascent/descent differences earlier in the text, as my earlier comments said, it is instructive, especially the decreased quality (increased sigmas) of the wind directions and speeds during ascent.

We would prefer to keep the analysis in this section. Differences for ascent and descent as well as the usability of comparing both measurement directions are mentioned in Section 2.6 and in lines 262, 665, 679 and Table 2.

However, the statement in lines 646-448 that "This indicates a possibly higher accuracy of the drone observations compared to operational airliner observations already assimilated into NWP models" must again be modified and weakened, since the limited study presented here used only a subset of altitude ranges and substantially limited the number of very high-speed wind reports that were available. Similarly, the specific humidity statistics are in improved in part because the drone samples were taken primarily at low temperature and therefore low saturation specific limits.

We changed the text (l 681) to:

*"For the descent profile, the average differences as well as the standard deviation of the differences between drone observations and radiosonde observations are in the range of (or even below) statistical measures shown in the study of Wagner et al., 2021, which includes a comparison of radiosonde data with observation data from the AMDAR/TAMDAR programme."*

Also, the authors seem focused on moisture in the middle-upper troposphere, saying that it is particularly important. I'll grant that moisture can have some influence at these levels, but the amount of moisture, its vertical structure and horizontal structure are much more important for forecasting/studying weather events from heavy precipitation and flooding to severe storms.

We are not sure which sentence the reviewer is referring to with the comment; therefore it is difficult to improve the text for making the statement clearer.

We agree that PBL observations are of superior importance for weather events, but think that the limited abilities of recent UAS platforms tended to disregard the advantages of sounding the middle-upper troposphere for weather event studies/forecast.

As a result of inadequacies like these throughout the text, it read still reads more like a preliminary progress report instead of a quantitatively verified scientific contribution to the literature. Although I support concept presented here, the paper is still not ready for publication, though a change in title would provide a major step toward defining the purpose of the effort.

We worked on the points that were raised, and in particular tried to make clearer the main intention, which is to introduce the achievement of a drone system capable of soundings up to 10 km altitude, with the potential to be used complementarily to radiosondes. We did not apply the perfect meteorological instrumentation for this, but this is left for the future.

Again, my intent here is to provide constructive suggestions for improving the paper and look forward to seeing a revised version.

We would like to thank the reviewer again for the detailed comments. We took into account most of them, but could not agree with every point. Including an uncertainty discussion was a huge effort, but also adds substantial value to the manuscript.

References:

Moninger, W. R., Benjamin, S. G., Jamison, B. D., Schlatter, T. W., Smith, T. L., & Szoke, E. J. (2010). Evaluation of Regional Aircraft Observations Using TAMDAR, *Weather and Forecasting*, *25*(2), 627-645. Retrieved Jan 10, 2023, from https://journals.ametsoc.org/view/journals/wefo/25/2/2009waf2222321_1.xml

---

## Author Response (AR3)

**Author Response Revision3 - amt-2022-236**

Drone-based meteorological observations up to the tropopause
Konrad Benedikt Bärfuss et al., Atmos. Meas. Tech. Discuss.
https://doi.org/10.5194/amt-2022-236, 2022

Please note: Author's response in blue color

**Report #1:**

As I said in my previous review, I am a strong supporter of the utility and benefit of automated aircraft reports to fill the time and space gaps left between other in situ observing systems. Upon rereading the article "Drone-based meteorological observations up to the tropopause a concept study" by Barfüss, Schmithüsen and Lampert, I feel that it is now ready for publication after a few minor changes. Some of my reasoning for this decision for the suggested changes is summarized below. I apologize for some possible disjointedness in the review, but that may in places reflect the need for more organization in the paper's organization.

To start, I thank the authors for their continued efforts to improve the paper. The current version is much more informative and leave many fewer points unresolved.

The Authors are grateful for your extensive reviews, which helped a lot for improving the article and additionally led to productive internal discussions. We highly appreciate your effort and would like to thank you.

Please note, that the line numbers correspond to the plain revised document (not the one with tracked changes).

The introduction remains essentially unchanged and includes very little discussion of problems that other authors have documented with automated reports from commercial aircraft that should provide a far more stable platform for collecting data and providing representative measurements or the atmosphere without being affected by artifacts related to aircraft stability. For example, AMDAR wind observations from longer-range aircraft are much more accurate (by a factor of 2) than TAMDAR reports obtained from smaller/lighter regional jets. Possibly more distracting to is the statement in the first sentence of the Abstract reads "with large data gaps in the atmospheric boundary layer, above the oceans and in polar regions", with only a brief mention of that "the feasibility of reaching an altitude of 10 km with a small meteorologically equipped drone is shown." Through the remainder of the text, the utility of the drone observations, however, is judged by the ability to reach 10km in a polar environment. The Abstract should be revised accordingly.

We re-wrote the abstract, now emphasizing more the main intention of the article, which is demonstrating the feasibility of reaching an altitude of 10 km with a drone on own propulsion. This provides a new tool for the community to gather data for weather forecast.

We changed the abstract text to:

"The main in-situ database for numerical weather prediction currently relies on radiosonde and airliner observations, with large systematic data gaps, horizontally in certain countries, above the

oceans and in polar regions, and vertically in the rapidly changing atmospheric boundary layer, but also up to the tropopause in areas with low air traffic.

These gaps might be patched by measurements with drones. They provide a significant improvement towards environment friendly additional data, avoiding waste and without the need for helium. So far such systems have not been regarded as a feasible alternative of performing measurements up to the upper troposphere.

In this article, the development of a drone system that is capable of sounding the atmosphere up to an altitude of 10\,km with own propulsion is presented, for which Antarctic and mid-European ambient conditions were taken into account: After an assessment of the environmental conditions at two exemplary radiosounding sites, the design of the system and the instrumentation are presented. Further, the process to get permissions for such flight tests even in the densely populated continent Europe is discussed, and methods to compare drone and radiosonde data for quality assessment are presented.

The main result is the technical achievement demonstrating the feasibility of reaching an altitude of 10\,km with a small meteorologically equipped drone using own propulsion. The first data are compared to radiosonde measurements, demonstrating an accuracy comparable to other aircraft based observations, despite the simplistic sensor package deployed. A detailed error discussion is performed.

The article closes with an outlook on the potential use of drones for filling data gaps in the troposphere."

Grammar and word choice continue to present problems throughout the paper. For example. In line 77, the term "Breakthrough requirements" is not defined. Are these "new' requirements or mesoscale requirements that were never intended to be fully met by GOS. Although lines 77-83 attempt to clarify this issue, inferring that 100 km horizontal and 1 km vertical resolution will not be sufficient to meet also local and regional needs, especially without a clear statement of temporal frequency.

Line 81: We clarified the term and added the missing information on observation cycle and timeliness.

In line 89, the term "back in the 60ies" should be "60s", or better should be simplified to "at that time". Also, in line 89, replacing "using binoculars" with "using theodolites" would suggest that the observations could be quantitative, not just qualitative.

Line 94: Switched to "at that time". We decided to stick with the word "binoculars" and clarified their use by adding "using binoculars to monitor the aircraft's attitude" which is not typically done with theodolites.

Lines 98-105 – The idea of using this application in the future is intriguing, but it must be noted that it will only have benefit if the full-resolution data can be downlinked and distributed in real time, which can't be done with the current system.

Line 111: We added "It must be noted here, that the deployment of drone systems for operational meteorology only has benefits, if data can be transferred and distributed in near real-time, which has not been demonstrated within most of the above-mentioned studies." to address this issue.

Line 122 – Add "extremely low specific humidities" to the list of challenges after "temperatures"

Line 131 – Replace "to play . . . and" with "as a future alternative to"

Line 156 – Replace "for a time period of" with "episodically over"

Line 167 – Be more direct at beginning of sentence by eliminating "trade-off" notion with "The LUCA system was designed . . ."

Line 171 – Replace 'Applying wind speed condition" with "Restricting wind speed conditions"

We changed the wording, thank you for your explicit suggestions!

Line 178 – Are you saying that LUCA will operate at times when radiosondes will fail, e.g., in conditions of "rainfall, snow, heavy turbulence and within clouds". No proof for that statement is shown, unless drones were launched alongside every radiosonde launch. This needs to be clarified.

Line 189:We added ",  but these capabilities have yet to be proven in upcoming measurement campaigns." to clarify the current state of testing.

Line 190 – The fact that 45% could severely reduce the number of drone profiles that are available, making the system appear to not be "all weather". Can you estimate how many LUCA flights would be missed due to the potential for icing?

Unfortunately not, as we did not find a theoretical estimation for the location of Neumayer III in Antarctica, and no in-situ data. In addition, UAS behave differently from manned aircraft in icing conditions, and trajectories may vary substantially, which could interact with the extent of the altitude band in which aircraft icing will occur.

Line 198 – Add "shown here" after "flights" and be clear whether an icing sensor was in fact installed, not just "prepared to be installed."

Line 211: We clarified that we did not install the icing sensor (but a camera instead) as the risk of icing was negligible during the demonstration flights presented in the article.

Line 220 – It would be good to include the typical ascent rate values along with the other performance specifications here so the reader doesn't have to hunt through the paper to find them

Line 233: Yes, we totally agree and added the missing specification.

Line 236 – What is the meaning of the first sentence? It could be read to imply that multiple sensors were used for each atmospheric parameter being measured. Eliminating it removes nothing of value from the text.

True – sentence eliminated.

Line 241 – Mention should be made both that the radiosonde sensors were designed to have a low ventilation rate as the balloon drifts with the wind and that a drone travelling at constant air speed

of 28 m/s will have much greater ventilation rate. How is these accounted for? If this is discussed later, say so.

Line 262:We revised the beginning of the section and included the statement: "The ventilation rate, which is assumed to be still higher than the ventilation rate of radiosonde sensors, minimises the response time of the sensors as the sensor is exposed to an increased amount of air per time compared to radiosondes."

I will stop with individual comments here but suggest that the authors review the text thoroughly for other wording errors or inconsistency, if any. Overall, the details of the presentation have improved significantly since the first version.

Lines 290-564 – The authors have provided a very good description of the data processing and error estimation procedures. While the information is extremely helpful, I found that the amount of detail presented distracted the reader from the flow of the paper's primary messages. As such, I highly recommend that this portion of the paper be moved to an appendix dedicated to the subject.

This fits well to our initial plan of the structure of our article, we highly appreciate the suggestion and moved the according sections to the appendix. The existence of the (from our point of view also very helpful) sections is indicated at the end of the introduction.

Line 334 – Are the coefficients the same during ascent and descent? Are these results consistent with documented hysteresis issues using radiosonde-like sensors for TAMDAR aircraft? Table 2 seems to imply that there is no hysteresis during ascent with a warm bias during descent, which is opposite of the TAMDAR results. How much is this a function of the choice of M ~ 0.07 in Equation 5?

Line 567: Yes, the coefficient m is the same for all flights – to clarify this, we included the phrase: "The coefficient $m$ which expresses the heat transfer rate from the fuselage to the sensor is regarded as independent of environmental conditions and universal for all temperature measurements with the drone \emph{LUCA}."

Hysteresis and observed bias should ideally be fully resolved (zero bias, no hysteresis) by the signal reconstruction and correction methods presented in the Appendix (besides the phase neutral averaging). This is not achieved with the coefficients and time constants applied to the data, which indicates the need for more measurements to calibrate/tune our post-processing parameters.

The influence of these coefficients and time constants fold in a complex way with the actual vertical profiles of temperature/humidity, so no simple function of hysteresis/bias and the coefficients/time-constants can be named here.

Line 451 – If the time constant for temperature reports is 21 seconds, please show how that translate into spatial and vertical averaging distances. E.g., does 21 seconds for a drone travelling at 28 m/s (and 10 m/s vertically) in still winds conditions equate to 0.588 km horizontal averaging distance and a 210 m vertical average (~21 hPa near the surface)? How would this change with tailwinds and headwinds?

Tailwinds and headwinds do not affect measurements of temperature and humidity.

As a time constant of a transfer function of type PT1 (control theory) has no representation in the time space without an explicit input series, the translation of the time constant value into spatial

distances is not valid. In addition, the assumed time constant of the sensor is recovered during the post-processing (signal reconstruction/inverse filtering), but indeed the applied phase neutral filter to smooth the results after signal reconstruction can be approximated to the width of a phase neutral mean filter (centred averaging).

Line 685: We added "Assuming a constant lapse rate of $-0.0065\,K\,m^{-1}$ and following the fundamentals of control theory, the signal reconstruction would correct for a hypothetical hysteresis of $1.38\,K$ during the ascent." and "[..]which is roughly comparable to a central average over a vertical extent of $120\,m$."

The horizontal averaging/smoothing is not relevant, as we climb in circles around an earth fixed point.

Line 460 – For RH, the question becomes more complicated. For the same still conditions, do3s this mean that the spatial and temporal smoothing at the surface are 0.420 km and 150 m (~15 hPa) respectively, which at 3 km elevation become1.68 km and 600 m (~40 hPa) and at 10 km are longer than the entire flight length (2000 s = 33.33 minutes), therefore making the RH values at high levels reflections of the average RH throughout the entire flight.

For raw measurements, this assumption is close to be right – but as we apply signal reconstruction, the effect is fully corrected in theory (assuming a perfect sensor model and zero noise). The simplified sensor model leads to errors, and noise in raw sensor measurements require applying a filter after the signal reconstruction (which itself relies on quasi-gradients of the measurement values and therefore amplifies measurement noise). Regarding this phase neutral filter to smooth out artefacts introduced by sensor noise as a central average with a variable window size is not any more intuitive, as the maximum theoretical vertical extent to be averaged exceeds 10 km.

---Please include vertical profiles of the horizontal and vertical averaging distances that have been applied to each type of observation somewhere in the text. Figure 11 might be a candidate location.

Figure 8: As stated before, we regard the horizontal averaging as not applicable for our trajectory. The vertical averaging is approximated for the temperature signal processing to a constant value of 120 m (stated in the caption to Figure 8), but we did not found a representative value for the humidity processing which supports readers in interpreting the results. We simply included the time constant for the phase neutral smoothing filter over altitude in the Figure (now Figure 8) and hope, this indicates the variable smoothing well for the audience.

Figure 5 and text that goes with it. – I like this example. Thanks for creating it.

Line 664 – Clearly state the data set details. As it stands, I don't know whether the statistics were obtained from the profiles in Fig. 9 or from all profiles in Fig. 8 until the second paragraph into the section.

Done in Line 383

Line 666 – The word "treats" is clearly not correct. Should this have been "threats" or better yet, "factors"?

Line 386: Changed to "factors".

Figure 10, center top panel (RH) – A shift of the apex of a probability density plot away from zero (in this case toward -2.5%) usually reflects a bias in an observing system. Please include a statement of how the variable time averaging that was applied in line 460 affected this.

Line 404: We mentioned  the possibly not finally tuned calibration parameters which would require a broader database with the phrase: "For the humidity measurement, the average of the differences between radiosonde and \emph{LUCA} measurements differs significantly between ascent and descent, indicating the possible need for further calibration and tuning of the post-processing parameters using a broader database in the future."

Table 2 – Please ad the number "6" before "ascent" and "descent" in the last sentence of the caption. It makes it easier for the reader to understand the sample size.

Numbers added.

End of paper – I think that it is still important not to ignore the issue of system and operations costs. That could be incorporated in the future, more extensive testing that was suggested. Because much of the thrust of the paper was addressed at substituting for radiosondes, lack of mention of the question weakens what has become a very good paper.

While we rather aim for supplementing radiosondes than substituting them, we added a few words in Line 510, but did not feel able to include explicit statements which address the financial aspect.

As I said in earlier reviews, I support concept presented here. With the enhancements that the authors have provided, the paper is now essentially ready for publication, after the small number comments are addressed. Again, my intent here has been to provide constructive suggestions for improving the paper. I do not need to see the final revised version.

I look forward to seeing the article in print – Ralph Petersen